# Subcellular localization of biomolecules and drug distribution by high-definition ion beam imaging

Xavier Rovira-Clavé[1,2,7], Sizun Jiang [1,2,7], Yunhao Bai[1,2], Bokai Zhu[1,2], Graham Barlow [1,2], Salil Bhate[1,2,3], Ahmet F. Coskun [1,2], Guojun Han [1,2], Chin-Min Kimmy Ho [1,2,5], Chuck Hitzman[4], Shih-Yu Chen[1,2,6,8], Felice-Alessio Bava[1,2,8] & Garry P. Nolan [1,2,8 ✉]

Simultaneous visualization of the relationship between multiple biomolecules and their ligands or small molecules at the nanometer scale in cells will enable greater understanding of how biological processes operate. We present here high-definition multiplex ion beam imaging (HD-MIBI), a secondary ion mass spectrometry approach capable of high-parameter imaging in 3D of targeted biological entities and exogenously added structurally-unmodified small molecules. With this technology, the atomic constituents of the biomolecules themselves can be used in our system as the "tag" and we demonstrate measurements down to ~30 nm lateral resolution. We correlated the subcellular localization of the chemotherapy drug cisplatin simultaneously with five subnuclear structures. Cisplatin was preferentially enriched in nuclear speckles and excluded from closed-chromatin regions, indicative of a role for cisplatin in active regions of chromatin. Unexpectedly, cells surviving multi-drug treatment with cisplatin and the BET inhibitor JQ1 demonstrated near total cisplatin exclusion from the nucleus, suggesting that selective subcellular drug relocalization may modulate resistance to this important chemotherapeutic treatment. Multiplexed high-resolution imaging techniques, such as HD-MIBI, will enable studies of biomolecules and drug distributions in biologically relevant subcellular microenvironments by visualizing the processes themselves in concert, rather than inferring mechanism through surrogate analyses.

[1] Department of Microbiology and Immunology, Stanford University, Stanford, CA, USA. [2] Department of Pathology, Stanford University, Stanford, CA, USA. [3] Department of Bioengineering, Stanford University, Stanford, CA, USA. [4] Stanford Nano Shared Facility, Stanford University, Stanford, CA, USA. [5] Present address: Institute of Plant and Microbial Biology, Academia Sinica, Taipei, Taiwan. [6] Present address: Institute of Biomedical Sciences, Academia Sinica, Taipei, Taiwan. [7] These authors contributed equally: Xavier Rovira-Clave, Sizun Jiang. [8] These authors jointly supervised this work: Shih-Yu Chen, Felice-Alessio Bava, Garry P. Nolan. ✉email: gnolan@stanford.edu

Interactions of molecular entities drive function in biological systems. When current understandings of cell function falter short of clinical utility, this drives a need for technologies that enable more precise analysis of cellular biomolecular components within cells in more native contexts. Development of multi-parameter high-resolution approaches that visualize relevant drugs and multiple cellular components at the molecular scale will be crucial to overcome limitations in our current understandings.

Fluorescence microscopy is currently the de-facto choice for biomolecular imaging, although multiplexing with fluorophores can be challenging due to the spectral overlap of fluorophores. Recent advancements using cyclic-hybridization protocols enable multiparameter visualization of proteins, RNA, and DNA[1–4], in part overcoming the spectral issue. These techniques have allowed a range of studies from single cells at super-resolution[5] to whole tissues[6] and have revealed novel insights on the spatial distribution of DNA and RNA in their biological context[5,7,8]. Acquisition time for imaging and sample positional shifts between cycles, which become more pronounced at super-resolution scales[9], places practical limits on the utility of these approaches. Often, imaging of small molecules, such as drugs, metabolites, or lipids, require the conjugation of chemical tags, and can vastly affect biological activities[10]. The ability to directly image atomic components as labels would, with sufficiently high resolution, allow for a three-dimensional (3D) reconstruction of the distribution of multiple biomolecules in whole cells. This will complement the tissue mapping efforts now ongoing worldwide by bringing subcellular maps of the 3D nucleome and proteome to more public use and application.

An alternative modality of biomolecular imaging is mass spectrometry imaging (MSI). Images are obtained via the rastering of a biological sample with a laser[11] or an ion beam[12] to liberate molecules to be analyzed by a mass spectrometer. We previously demonstrated the ability of secondary ion beam imaging (SIMS), a form of MSI, to image multiple targeted-proteins in tissue samples using multiplexed ion beam imaging (MIBI)[13]. In MIBI, proteins of interest are stained with isotope-tagged antibodies, and the sample is rasterized with an oxygen-based primary ion beam to generate a two-dimensional composite image of up to 50 proteins, down to 260 nm resolution[14]. A similar approach, imaging mass cytometry (IMC), allows protein and RNA to be visualized in tissues using laser ablation at 1000 nm resolution[15,16]. Although MIBI can define subcellular distribution of dozens of biomolecules, resolution limits (determined by the ion gun, vibration effects, electromagnetic lenses, and the working distance from the gun to the sample) are currently the "filters" that define the limits for 3D cellular imaging at the nanoscale. Instrument development in the SIMS field has yielded devices that enable 3D visualization of small molecules with subcellular resolution, suggesting the potential for multiplexed analysis of dozens of biomolecules and small molecules at the nanoscale[17–24]. Also, analysis of specific single proteins via SIMS at high lateral resolutions has been previously achieved[25–27], but they lack the specific and scalable targeting of multiple biomolecules.

Here, we report the application of HD-MIBI using a positively charged cesium primary ion beam that allows visualization of subcellular structures with lateral (XY) and axial (Z) resolutions down to ~30 and 5 nm, respectively. We highlight the development of a toolkit for specific protein imaging using multiple isotope-tagged antibodies. HD-MIBI can be extended to determine the precise subcellular locations of multiple small molecules, proteins, and nucleic acids.

We demonstrate the capabilities of HD-MIBI by simultaneously imaging five distinct subnuclear structures and the chemotherapeutic drug cisplatin. Cisplatin was approved in the US in 1978 for testicular and ovarian cancer and remains a key first-line chemotherapeutic across a range of cancers. Cisplatin has the ability to cross-link to guanines or adenines in DNA, thereby interfering with cancer cell replication or gene function[28]. Utilizing a computational framework incorporating dimensional reduction and clustering methods to analyze HD-MIBI data, we identify microenvironments within the nucleus, which we term nuclear neighborhoods, wherein cisplatin shows non-uniform distributions. Notably, one discovered neighborhood wherein cisplatin was enriched was the nuclear speckle[29]. Cisplatin treatment led to aberrant RNA processing, consistent with previous descriptions[30]. Surprisingly, multi-drug treatment of cisplatin and the BET inhibitor JQ1 revealed near complete exclusion of cisplatin from nearly all nuclear neighborhoods in resistant cells. HD-MIBI enables in situ imaging of a drug in its native stage to highlight how cancer cells render themselves resistant to cisplatin treatment through an active exclusion mechanism, suggesting how anti-resistance modalities can be developed to overcome cancer cell resistance to cisplatin. This exemplifies how direct observation of small molecules, co-localized with relevant multi-component biomolecular entities, opens new avenues to understand multiple new mechanisms whereby cells are both susceptible to, and adapt to, drug partitioning in cells.

## Results

**Overview of HD-MIBI**. HD-MIBI uses a positively charged cesium primary ion beam with a small spot size to obtain pseudo 3D super-resolution multiparametric visualization of cellular features (Fig. 1a), including the distribution of small molecules. Cells are cultured on a conductive silicon substrate and treated with cisplatin, a small molecule containing a platinum atom that can be efficiently ionized by the cesium beam. Cells are then fixed, simultaneously stained with multiple isotope-tagged antibodies using a protocol optimized for intracellular staining, dried under vacuum, and loaded into the SIMS instrument. To preserve the fine cellular structures under high vacuum, cell preparation and imaging conditions are nearly identical to that in standard scanning electron microscopy. A cell of interest is identified with an optical camera and then iteratively rasterized with the cesium primary ion beam. This process releases a cloud of negative secondary ions at the point of contact on the cell surface, which are collected and recorded pixel-by-pixel using a magnetic sector mass spectrometer. This results in a two-dimensional (2D) image for each analyzed isotope. Serial acquisition of hundreds of planes yields a stack of 2D images for each isotope that can be merged to obtain a multiparameter visualization of the cellular features of interest, and volumetrically reconstructed in 3D at resolutions comparable to other super-resolution fluorescence-based methods. Application of dimensionality reduction and clustering methods to analyze HD-MIBI data enables identification of subcellular microenvironments within cells.

**Design and validation of an antibody tagging strategy for HD-MIBI**. In our original MIBI approach, an oxygen duoplasmatron source was used to generate 2D images of tissue sections stained with lanthanide-tagged antibodies down to 260 nm resolution[13]. Attaining resolutions beyond this limit requires a beam source physically capable of forming a tighter beam. The cesium primary ion beam can achieve resolutions down to 30 nm in biological samples[17,24,31], but cannot be used in conjunction with lanthanide-tagged antibodies due to inefficient ionization[32]. To circumvent this issue, we developed a new labeling strategy that harnesses the ionization efficiency of the cesium beam, including that for halogen elements. DNA is a versatile polymer that has been extensively functionalized for imaging purposes[3,33–35]. We

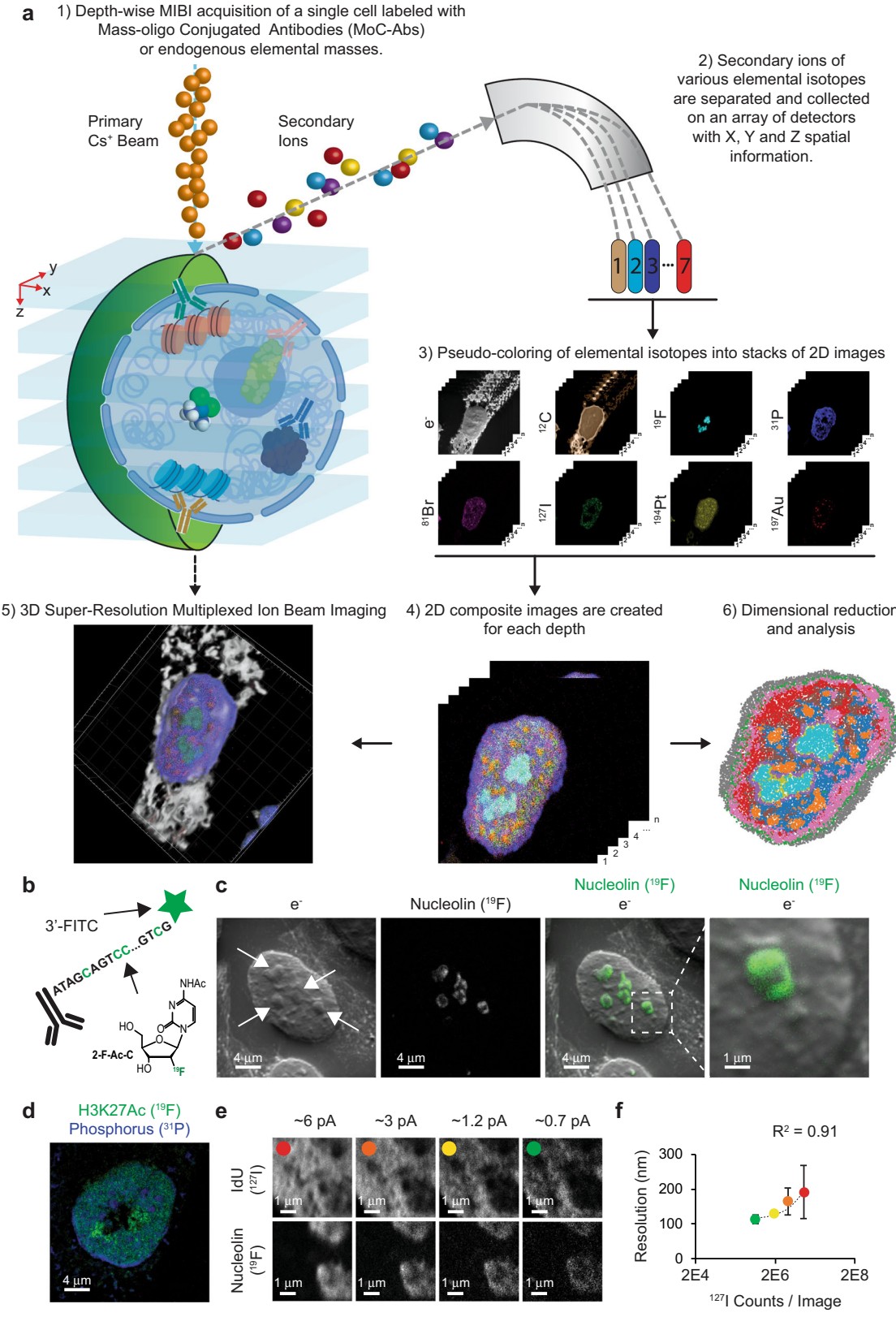

reasoned that antibodies conjugated with single-stranded DNA (ssDNA) oligonucleotides carrying stable isotopes, such as halogens not normally abundant in cells, would enable super-resolution protein visualization by MIBI. To this end, we developed the mass-oligonucleotide-conjugated antibody (MoC-Ab). Isotope-modified ssDNA oligonucleotides were conjugated to antibodies through thiol-maleimide click chemistry (Fig. S1)[36]. MoC-Abs can be used for intracellular protein staining in an optimized protocol that involves paraformaldehyde fixation, permeabilization, and blocking with a high-salt concentration in the presence of sheared salmon sperm DNA to avoid non-specific binding (Fig. S2).

**Fig. 1 Super-resolution visualization of nuclear structures using HD-MIBI. a** Workflow of super-resolution ion beam imaging (HD-MIBI). (1) Cells are treated with the drug of study, fixed, and stained with MoC-Abs. Endogenous elemental masses can also be detected. (2) Cells are rasterized by a cesium primary ion beam, and the secondary ions are collected by a magnetic sector mass spectrometer. (3) Spatial information is serially recorded in up to 8 channels simultaneously. (4) A composite image of the cell is reconstructed by combining and pseudo-coloring the total number of channels at each depth. (5) Serial acquisitions via HD-MIBI yields a depth profile that reveals the specific subcellular localization of endogenous and labeled targets. A 3D rendering of the depths are shown here. (6) Ion count extraction on a pixel-by-pixel basis for feature extraction allows the application of computational methods to identify subcellular interactions, such as nuclear neighborhoods. **b** Schematic of $^{19}$F-based MoC-Ab. The oligonucleotide includes a FITC fluorophore at the 3′ end and the modified base 2-F-Ac-C (green) in place of deoxycytidine. **c** Representative $^-$e and $^{19}$F images of a HeLa cell stained with anti-nucleolin-$^{19}$F/ FITC. From left to right: (1) $^-$e image of a HeLa cell; nucleoli are indicated with white arrows. (2) Image based on $^{19}$F signals originating from anti-nucleolin-$^{19}$F/FITC. (3) Overlay of images showing the co-localization of $^{19}$F signal with the subnuclear structures predicted to be the nucleoli from the $^-$e image. (4) An enlarged image of a nucleolus. $n = 3$. **d** Representative HD-MIBI image of a HeLa cell stained with anti-H3K27Ac-$^{19}$F/FITC. Overlay of ion images for $^{19}$F (green) and $^{31}$P (blue). The image is the sum of 10 consecutive planes. $n = 3$. **e** HeLa cells were labeled for 24 h with IdU ($^{127}$I) and stained with anti-nucleolin-$^{19}$F/FITC. Using HD-MIBI, 40 depths of a single cell were acquired with an increase of current every 10 scans. Images of newly synthesized DNA (top) and nucleolin (middle) and their overlay (bottom) show details of a region within the nucleus, at different currents. Details of DNA and nucleolus in these settings are increasing by lowering the current, at the cost of ion yield. Colored dots indicate the image used for the quantification shown in panel (**f**). $n = 3$. **f** Quantification of the relationship between HD-MIBI resolution and secondary ion counts. The lateral resolution of $^{127}$I (present in IdU) at different currents from panel (**e**) was calculated using the 84–16% criterion. The $R^2$ for the relationship between the increase in resolution and decrease in ion yield was 0.91. Data are presented as mean values +/−SD. $n = 3$ line scans per cell per condition. See Fig. S6 for details.

In a proof-of-concept experiment, the single stable fluorine isotope ($^{19}$F) was incorporated into a FITC-labeled ssDNA by substituting all cytidines with the commercially available N4-acetyl-2′-fluoro-2′-deoxycytidine (2-F-Ac-C) (Fig. 1b; $^{19}$F/FITC MoC-Ab). The behavior of MoC-Abs is similar to that of unmodified antibodies as shown by confocal microscopy analyses of target specificity, staining intensity, and background levels of nucleolin staining in HeLa cells (Fig. S3A-C). We next assessed the performance of MoC-Abs in HD-MIBI by visualizing HeLa cells stained with anti-nucleolin-$^{19}$F/FITC. Rastering a HeLa cell with the cesium ion beam revealed an enriched $^{19}$F signal inside the nucleus akin to the fluorescence signal observed using confocal microscopy (Fig. S3D, right images). Moreover, control experiments without anti-nucleolin-$^{19}$F/FITC resulted in an absence of $^{19}$F signal (Fig. S3D, left images), suggesting that the $^{19}$F signal observed in the stained cell resulted from the MoC-Ab. The high $^{31}$P content of the DNA phosphate backbone enabled visualization of the nucleus in the HD-MIBI experiment in a manner analogous to use of DAPI in conventional fluorescent microscopy (Fig. S4).

In HD-MIBI, secondary electron ($^-$e) and elemental ion information are simultaneously recorded for each plane. The $^-$e image, akin to that from a scanning electron microscope, enables the identification of certain subnuclear structures such as the nucleolus (Fig. 1c, left image). To confirm the specificity of anti-nucleolin-$^{19}$F/FITC, we merged the $^{19}$F and $^-$e signals and observed a specific enrichment of the $^{19}$F signal in nucleolar structures identified through the $^-$e image (Fig. 1c, overlay and right image). To demonstrate the versatility of HD-MIBI, we labeled transcriptionally active chromatin using a $^{19}$F/FITC MoC-Ab that binds to H3K27Ac (Fig. 1d). The $^{19}$F signal overlapped with nuclear regions low in $^{31}$P confirming the specificity of anti-H3K27Ac-$^{19}$F/FITC (Fig. S5). These results demonstrate that detection of MoC-Abs labeled with $^{19}$F by ion beam imaging can reveal subnuclear structures.

In SIMS, the beam size dictates the lateral resolution and ion yield of the acquired image; the size of the beam is proportional to its current[37]. We quantified the effect of beam current on the ion count per pixel in our samples by acquiring sequential images of the nucleolus of a HeLa cell. Raster size, pixel dwell time, and number of pixels acquired were maintained at constant levels, while the beam current was varied (Fig. S6A, B). We observed that the increase in current scaled with ion counts per pixel (Fig. S6C, D) at the expense of resolution (Fig. S6B). To quantify the lateral resolution, we acquired sequential images of a HeLa

cell incubated for 24 h with 5-iodo-2′-deoxyuridine (IdU) to specifically label newly replicated DNA. Under the conditions tested, reduction of the beam current improved the lateral resolution from 193 nm to 113 nm and lowered the total ion count 16-fold ($R^2 = 0.91$; Fig. 1e, f and Fig. S6E-I). These results show that HD-MIBI resolves targeted subcellular structures with a resolution, to our knowledge, not attainable with current multiplexed MSI methods that target specific proteins[13,15].

**Multiparametric imaging and unsupervised identification of subnuclear structures.** NanoSIMS, the instrument we used for HD-MIBI in these experiments, has a five-log dynamic range that enables simultaneous detection of rich and trace elements[13]. MoC-Abs contains 48 to 72 isotopic labels per antibody, resulting in a limit of detection of down to three MoC-Abs per voxel in certain analytical conditions (Supplementary Note 1). The limit of detection for each element depends on individual ionization efficiencies driven by the cesium ion beam as characterized systematically previously[38].

We leveraged the capabilities of mass spectrometry for simultaneous data acquisition by expanding on the number of elements used to label MoC-Abs. Halogens are efficiently ionized by the cesium beam due to their high electronegativity and were the elements of choice in the experiments herein. Bromine ($^{79/81}$Br) and iodine ($^{127}$I) are two halogen tags used previously to covalently linked nucleotides for cell proliferation, DNA replication, and RNA synthesis studies, making them biologically validated tags to be harnessed for HD-MIBI. We synthesized and validated MoC-Abs containing 5-bromo-2′-deoxycytidine (5-Br-dC) and 5-iodo-2′-deoxycytidine (5-I-dC) for the labeling of centromeres (anti-CENP-A-$^{81}$Br/Cy3) (Fig. 2a and Fig. S7) and transcriptionally active chromatin (anti-H3K27Ac-$^{127}$I/Cy5) (Fig. 2b and Fig. S8), respectively. To further demonstrate the specificity of MoC-Abs, we labeled transcriptionally silent chromatin (anti-H3K9me3-$^{81}$Br/Cy3) (Fig. S9A) and nucleoli (anti-nucleolin-$^{127}$I/Cy5) (Fig. S9B) in HeLa cells, obtaining the expected patterns for each of these reagents. To exemplify additional capabilities of HD-MIBI other than identification of nuclear structures in cultured cells, we successfully imaged mitochondria in HeLa cells (anti-TOM20-$^{81}$Br/Cy3) (Fig. S9C), and transcriptionally silent chromatin in frozen tissue sections (anti-H3K9me3-$^{81}$Br/Cy3) (Fig. S10).

In addition to nucleotide-based methods, other reagents can also be harnessed for high-resolution imaging of specific proteins using SIMS[25–27]. To demonstrate this versatility, we stained HeLa

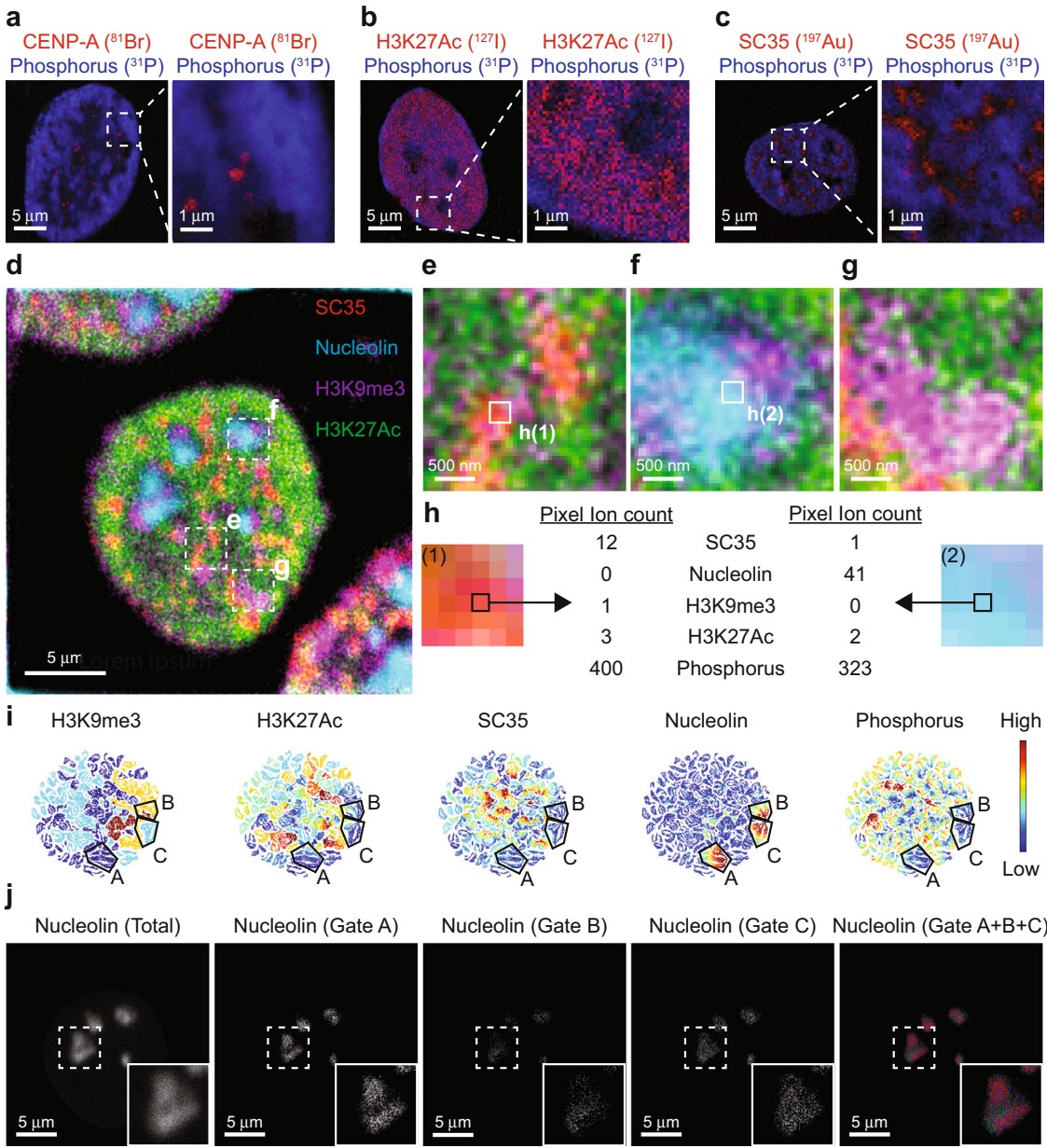

**Fig. 2 Multiparametric HD-MIBI using MoC-Abs identifies unique subcellular features. a–c** (Left) Representative HD-MIBI images of a HeLa cell stained with **a** anti-CENP-A-$^{81}$Br/Cy3, **b** anti-H3K27Ac-$^{127}$Ir/Cy5, and **c** anti-SC35-biotin recognized by streptavidin-$^{197}$Au/FITC. Overlay of ion images for MoC-Ab (red) and phosphorus ($^{31}$P; blue). (Right) A digital zoom of boxed area in the original image to show specificity of MoC-Ab signal. All images consist of the sums of 10 consecutive planes. $n = 3$. **d** Composite HD-MIBI image of nucleolin (cyan), H3K9me3 (magenta), H3K27Ac (green), and SC35 (red) in a HeLa cell. Cells were stained with anti-nucleolin-$^{19}$F/FITC, anti-H3K9me3-$^{81}$Br/Cy3, anti-H3K27Ac-$^{127}$Ir/Cy5, and anti-SC35-biotin (recognized by streptavidin-$^{197}$Au/FITC). The image consists of the sum of 10 consecutive planes. $n = 3$. **e–g** Multiple enlarged images from boxed regions in panel (**d**) showing **e** a nuclear speckle, **f** a nucleolus, and **g** heterochromatin. **h** The largest nucleus in panel (**d**) was masked using the phosphorus signal. The ion counts for each acquired marker (nucleolin, $^{19}$F; DNA, $^{31}$P; H3K9me3, $^{81}$Br; H3K27Ac, $^{127}$I; and SC35, $^{197}$Au) were extracted. **i** t-SNE maps of the nucleus in the center of the image shown in panel (**d**) were created using the ion counts for each of the 84,836 pixels from the mask described in (**h**). The ion count for each marker in each pixel was min–max normalized to scale the range to [0, 1]. Each point represents a pixel. Pixels grouped in distinct regions based on the expression of each marker. The color in each map represents the intensity of the indicated marker in each pixel. The areas marked A–C indicate three distinct groups of nucleolin-positive pixels identified manually in the nucleolin map. **j** Grouped pixels from the unsupervised viSNE map are differentially distributed in space as shown in HD-MIBI images of (1) total nucleolin ($^{19}$F) from the HeLa cell shown in panel (**d**), (2) $^{19}$F signals from pixels within gate A, (3) gate B, (4) gate C, and (5) overlay of $^{19}$F signals from gates A (red), B (green), and C (blue).

cells with phalloidin conjugated to ATTO514, a fluorophore containing six $^{19}$F atoms (Fig. S11A). The distribution of actin filaments observed in HeLa cells labeled with this reagent was comparable in confocal microscopy and HD-MIBI images (Fig. S11B, C). Nuclear speckles are nuclear domains containing inter-chromatin material enriched in pre-mRNA splicing

components[29]. We also validated detection of SC35, a protein enriched in nuclear speckles, using anti-SC35-biotin with streptavidin-labeled 1.4 nm gold nanoparticles (streptavidin-$^{197}$Au/FITC) by HD-MIBI (Fig. 2c and Fig. S12). We further validated this tool by labeling nucleoli (anti-NPM1-biotin recognized by streptavidin-$^{197}$Au/FITC) (Fig. S13). Collectively,

these data demonstrate that distinct atoms and tools are suitable for specific protein imaging using HD-MIBI.

We observed minimal crosstalk between channels as defined by isotope signal quantification in unstained channels of HeLa cells stained with MoC-Abs, one at a time (Fig. S14); therefore, multiplexed imaging, in a manner akin to fluorescence microscopy, can be performed using HD-MIBI (Fig. S15). To demonstrate the multiparameter capabilities of HD-MIBI, we simultaneously visualized nucleolin, H3K9me3, H3K27Ac, and SC35 in HeLa cells (Fig. 2d). Magnification of the composite image revealed a complexity unable to be concluded by simple observations of single-channel images (Fig. 2e–g and Fig. S16). We extracted the ion count per channel on each pixel of the multiplexed image (Fig. 2h) and used t-distributed stochastic neighbor embedding (t-SNE) to identify subnuclear structures (Fig. 2i)[39]. Pixels with high expression of nucleolin were manually grouped into three distinct regions that had variable levels of H3K9me3 and low levels of SC35, H3K27Ac, and phosphorus (Fig. 2i; gates A to C).

To highlight diversity in nucleolar organization, we categorized the pixels within nucleolin-rich regions in the three distinct subsets and generated a composite image of spatial distribution (Fig. 2j). We observed that the pixels with high (Fig. 2j, gate B) or low (Fig. 2j, gate A) H3K9me3 signal occupy contrasting locations within the nucleolin-rich regions of this cell. Thus, HD-MIBI can be used to visualize multiple isotopes simultaneously, and the application of dimensionality reduction techniques to the data enables the identification of distinct subnuclear structures.

**Pseudo 3D reconstruction at the nanoscale with HD-MIBI.** In HD-MIBI, ion beam current and residence time at each spot can yield secondary ion information for each plane that results from the ablation of just a few nanometers of the sample surface. Having observed that sequential planes have similar ion counts (Fig. S17), we reasoned that whole 3D nuclear reconstruction could be achieved with HD-MIBI if data were acquired on numerous consecutive planes. Axial resolution at a given analytical condition can be inferred by quantifying how many depths are required to erode a structure or a sample of a known thickness[40] (Fig. S18A-C). Previous axial resolution quantifications in biological samples on the nanoSIMS were determined to be around 5 nm[41]. In agreement with this measurement, reconstructions of centromeres from HD-MIBI (Fig. S18D) resembled those identified by recent super-resolution microscopy experiments[42].

As a preliminary experiment, and to demonstrate the feasibility of whole organelle pseudo 3D reconstruction, we acquired 785 planes of a single nucleolus (Fig. 3a and Fig. S19) and performed a volumetric reconstruction (Fig. 3b and Movies S1 and S2). The 3D nucleolar shape is different from other cellular substructures, such as nuclear speckles or mitochondria (Fig. S20). HD-MIBI can resolve juxtaposed subnuclear structures since a 50-plane acquisition was sufficient to distinguish between the centromere, nucleolus, and nuclear speckles in a HeLa cell (Fig. 3c). As further validation, we performed 3D rendering of 400 planes from a single HeLa cell labeled with anti-nucleolin-$^{19}$F, anti-H3K9me3-$^{81}$Br, anti-H3K27Ac-$^{127}$I, and anti-SC35 (recognized by streptavidin-$^{197}$Au), creating a composite 3D visualization of the nucleus (Fig. 3d and Movie S3). These results indicate how HD-MIBI enables whole-cell reconstruction of the genome and epigenome. Combined with an analytical framework, as shown below, HD-MIBI allows for intricate dissection of subcellular microenvironments in single cells.

**Iterative HD-MIBI resolves nanoscale nucleolar components at the molecular scale.** A lateral resolution down to 30 nm in biological samples has been previously reported using the cesium primary ion beam, by focusing it to an area of interest measuring a few microns[17,31]. During each pass of the primary ion beam, only the top material contacted is ablated. Most of the sample is preserved due to the high axial resolution (~5 nm) and thus is available for further imaging. We reasoned that we could perform an initial acquisition of a target cell to define a region of interest (ROI) and subsequently focus on that subcellular feature for higher resolution profiling (Fig. 3e, workflow). Indeed, re-probing a region of interest in an IdU-treated cell enabled iterative imaging at higher resolution (Fig. 3e, IdU images).

To achieve our highest benchmarked resolution in MSI of specific proteins, we performed iterative HD-MIBI in a single HeLa cell labeled with anti-nucleolin-$^{19}$F. Using line scans across the image to quantify the distance between $^{19}$F signals as commonly done for fluorescence-based super-resolution images[43], we identified nucleolin molecules spaced at about 30 nm apart (Fig. 3f and Fig. S21). This conforms with an expected size of a nucleolin monomer of ~76 kDa, with an estimated globular diameter of about 5–10 nm, and would be expected to interact with several other proteins in the nucleolar complex in its varied roles as a nucleosomal remodeling agent, amongst other activities[44]. The secondary electron image of a cell before and after an iterative HD-MIBI acquisition showed a generally even ablation of the field of view (FOV), which focused on three nucleoli (Fig. 3g, e⁻ images). A 3D reconstruction of a single nucleolus from this acquisition (Fig. 3g) revealed the expected spatial distribution of nucleolar proteins within a single nucleolus (Fig. 3g and Fig. S22). Hence, a first scan to define an area of interest and subsequent imaging to enhance the resolution revealed molecules spaced about 30 nm apart.

**Nuclear neighborhoods enable the identification of regions of drug enrichment.** SIMS enables subcellular visualization of exogenously incorporated small molecules and has been previously used to study the cellular distribution of the metallo-drug cisplatin[45], which is used to treat various types of cancer[46]. The presumed mechanism of cisplatin action is cross-linking to DNA, formation of adducts, and interference in DNA replication. The platinum atom of cisplatin (Fig. S23A) can be efficiently ionized using a cesium beam in SIMS[47], providing an ideal drug that can also function natively as a tag. We incubated TYK-nu cells, an immortalized ovarian cancer cell line, for 24 h with a range of concentrations of cisplatin to identify a concentration at which drug was internalized but caused minimal cell death (Fig. S23B). Naturally, occurring platinum is composed of five stable isotopes (Fig. S23C); we took advantage of isotopically pure cisplatin ($^{194}$Pt) to maximize Pt counts on a single SIMS channel. Cells treated with 15 μM cisplatin ($^{194}$Pt) for 72 h exhibited $^{194}$Pt counts well above the background (Fig. S23D), with signal observed in cytoplasm and nuclei (Fig. 4a), consistent previous SIMS data[45]. Minimal crosstalk between mass channels was observed (Fig. S23E). We performed MoC-Ab staining of cisplatin-treated cells and acquired images of ⁻e, $^{12}$C (indicative of overall cellular structure), $^{19}$F (nucleolin), $^{31}$P (DNA), $^{81}$Br (H3K9me3), $^{127}$I (H3K27Ac), $^{194}$Pt (cisplatin), and $^{197}$Au (SC35) to generate an eight-channel image at super-resolution. HD-MIBI revealed a non-uniform distribution of cisplatin in TYK-nu nuclei, and composite images enabled the visualization of the subnuclear distribution of cisplatin (Fig. 4a).

Biomolecules in the nucleus are organized spatially to enable proximity-determined pockets of interaction[48,49], akin to tissue[3,14,50] and subcellular microenvironments[51,52]. In such a

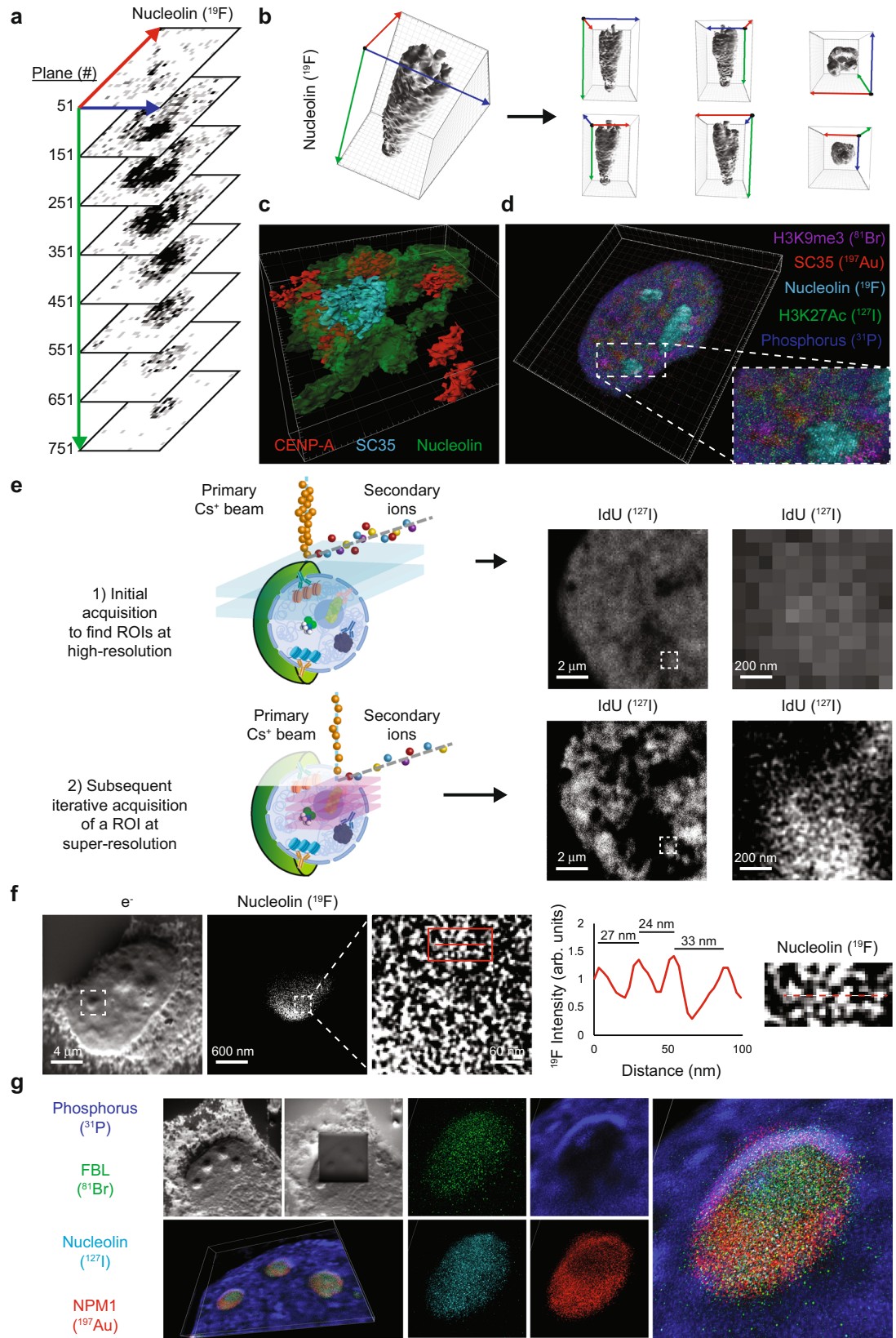

nuclear neighborhood, the function of various biomolecules might be dependent on their spatial localization and the context of other nearby constituents. We used the multiplexed data generated from HD-MIBI to identify distinct nuclear neighborhoods. First, whole-cell HD-MIBI acquisition was performed on cisplatin-treated cells (Fig. 4b, top left, and Movie S4). We then applied a sliding window, consisting of $3 \times 3 \times 10$ pixels, to extract the average isotope counts per window for five channels (H3K9me3, H3K27Ac, SC35, nucleolin, and phosphorus; Fig. 4b, top right). Unsupervised clustering was then performed on these extracted features to group windows based on similarity (Fig. 4b, bottom right). Individual voxels were then pseudo-colored based

**Fig. 3 3D nanoscale imaging of the nucleus through iterative HD-MIBI. a** Representative single-plane HD-MIBI images at different depths of a HeLa cell nucleolus. HeLa cells were stained with anti-nucleolin-$^{19}$F/FITC, and 785 individual planes were acquired to obtain HD-MIBI images of a nucleolus from its appearance to its disappearance. Single planes every 100 depths show a distinctive molecular distribution of nucleolin in the 3D space. See Fig. S19 for images of each individual plane. Blue, red, and green arrows indicate x-axis, y-axis, and z-axis, respectively. **b** (Left) 3D surface reconstruction of images of nucleolin staining of a nucleolus shown in panel (**a**). (Right) Overviews of the same nucleolus along x-axis (blue arrow), y-axis (red arrow), and z-axis (green arrow) with the origin represented as a black dot. **c** Representative 3D surface reconstruction of nucleolus (green), centromeres (red), and nuclear speckles (cyan). HeLa cells were stained with anti-nucleolin-$^{19}$F/FITC, anti-CENP-A-$^{81}$Br/Cy3, and anti-SC35-biotin (recognized by streptavidin-$^{197}$Au/FITC). The image consists of the 3D reconstruction of a stack of 40 consecutive planes. **d** Representative 3D reconstruction of nucleolin (cyan), phosphorus (blue), H3K9me3 (magenta), H3K27Ac (green), and SC35 (red) in a HeLa cell stained with anti-nucleolin-$^{19}$F/FITC, anti-H3K9me3-$^{81}$Br/Cy3, anti-H3K27Ac-$^{127}$I/Cy5, and anti-SC35-biotin (recognized by streptavidin-$^{197}$Au/FITC). The image consists of the 3D reconstruction of a stack of 400 consecutive planes. Enlarged images from the boxed region show details of marker distribution. **e** (Left) Workflow of iterative HD-MIBI: (1) Five to ten depths at high current are acquired in a cell of interest at 25 × 25 μm to identify a ROI. (2) Iterative acquisition is performed by focusing the beam into the ROI at lower currents in a smaller area of 5 × 5 μm or 10 × 10 μm. (Right) (Top) Representative region of the IdU signal ($^{127}$I) in a nucleus of a HeLa cell labeled for 24 h with IdU ($^{127}$I). (Bottom) A 10 × 10 μm ROI was acquired for super-resolution imaging of chromatin folding. Enlarged image of the boxed region shows fine detail of IdU labeled chromatin. n = 2. **f** Quantification of the resolution of iterative HD-MIBI imaging of a nucleolus. From left to right: (1) ⁻e image of a HeLa cell reveals subnuclear structures including the nucleoli chosen for iterative HD-MIBI. (2) A 3 × 3 μm ROI was acquired by iterative HD-MIBI for nucleolin ($^{19}$F). n = 2. (3) An enlarged view from the boxed region in image 2 showing the nanoscale organization of nucleolin ($^{19}$F). (4) Line scan along the red line in the boxed region in image 3 demonstrates identification of molecules spaced about 30 nm. (5) An enlarged view of the boxed region in image 3. See Fig. S21 for additional examples. **g** Iterative HD-MIBI 3D reconstruction. HeLa cells were stained with anti-FBL-$^{81}$Br/Cy3, anti-nucleolin-$^{127}$Ir/Cy5, and anti-NPM1-biotin (detected with streptavidin-$^{197}$Au/FITC). Iterative HD-MIBI was performed on a site with three nucleolin. The ⁻e image confirmed that the ROI was acquired. Nucleolin (cyan), phosphorus (blue), FBL (green), and NPM1 (red) were used for 3D reconstruction from 40 consecutive planes. These images identify the granular component (GC, NPM1-positive), dense fibrillar component (FBL- and nucleolin-positive), and perinucleolar heterochromatin (PNC, phosphorus-high).

on these groups for visualization purposes (Fig. 4b, bottom left). In total, 20,000 randomly sampled 3D voxels from the middle 40 Z-planes were extracted from each of the 2 cells, for a sum of 40,000 voxels.

Clustering of these 40,000 voxels based on the five channels gave rise to 10 distinct clusters, which represented 10 different nuclear neighborhoods (Fig. 4c) at this scale of observation and with these markers. We used the histone markers H3K9me3 and H3K27Ac as well-characterized surrogates for heterochromatin (higher DNA density) and euchromatin (low DNA density), respectively[53]. That these markers were consistently located in separable clusters supports this unsupervised approach to distinguish fine structure details (Fig. 4c, neighborhoods 5, 7, and 10). For instance, regions that are high in H3K9me3 also had abundant phosphorus, as expected because of the condensed nature of heterochromatin (Fig. 4c, neighborhood 5). We colored the voxels in each nucleus based on these groups and projected them onto a 2D plane to better visualize the spatial distribution of the nuclear neighborhoods (Fig. 4d). These images reveal distinctive nuclear structures, such as nucleoli (neighborhood 1), nuclear speckles (neighborhood 4), and chromatin that resembles lamin-associated domains (neighborhood 8). Features that are difficult to distinguish visually in the HD-MIBI images (Fig. 4a) could now be appreciated as neighborhood groupings of molecules (Fig. 4d), such as perinucleolar heterochromatin (neighborhood 9), as well as heterochromatin-like regions (neighborhood 10), and euchromatin (neighborhoods 6 and 7). This is also able to distinguish features that appear to be edge effects (neighborhoods 2 and 3)—note that interface regions themselves are not always artifacts and can be used to denote domain boundaries or even interaction regions of mixed molecules.

Identification of subcellular microenvironments, combined with the spatial characterization of atomic components from small molecules, offered the ability to create subcellular maps for multiple types of biomolecules and to determine the affinity of cisplatin for these different nuclear neighborhoods. In the two cells analyzed in 3D at ~150 nm axial and ~5 nm axial resolution (each cell takes >20 h), we observed that nuclear speckles contained high levels of cisplatin (Fig. 4e, neighborhood 4).

Cisplatin was lower in heterochromatin regions and lamin-associated domains than in other regions (Fig. 4e, neighborhoods 5 and 8).

These results validated HD-MIBI as a tool allowing study of the molecular scale localization of small molecules and showed a uniquely differential distribution of cisplatin into distinct nuclear neighborhoods.

**Identification of nuclear neighborhood interactions leads to underscore a role for cisplatin in promoting aberrant RNA maturation.** To better observe the spatial enrichment of cisplatin in the nuclear speckles relative to their depletion from closed chromatin, we created super-resolution images of cisplatin localization using iterative HD-MIBI (Fig. 5a and Fig. S24). Diversity of nuclear organization is appreciated upon inspection of iterative HD-MIBI images (Fig. 5a, right panels). We combined the visual nature of dimensionality reduction methods with the spatial information encoded within our data. This was initiated by randomly selecting 20,000 voxels across five different cells for combined analysis (see "Methods"). We then performed dimensional reduction using t-SNE on the log-transformed, mean isotope counts per voxel (Fig. S25A). As expected, signals were grouped together based on levels of marker expression (Fig. S25A). We next applied unsupervised hierarchical clustering followed by manual annotation of the clusters. This led to the identification of 11 unique nuclear neighborhoods as demarked (Fig. 5b–d and Fig. S25B-D).

Neighborhoods 1 and 2 were enriched for the heterochromatin marker H3K9me3, while neighborhoods 3 and 4 were enriched for the euchromatin marker H3K27ac. Neighborhood 5 represented cisplatin present in regions of euchromatin, while neighborhoods 6 and 7 represented cisplatin present in regions containing SC35 that either contained active chromatin (i.e., high in H3K27ac, SC35, and phosphorus) or inactive chromatin (low in H3K27ac, high in SC35 and phosphorus). Neighborhoods 8 and 9 represent well-recognized nuclear regions, rich in SC35 and Nucleolin, respectively, while neighborhood 10 indicated cisplatin-rich regions of the nucleolus. Each neighborhood was well represented in each cell (Fig. S25E). We next reconstructed

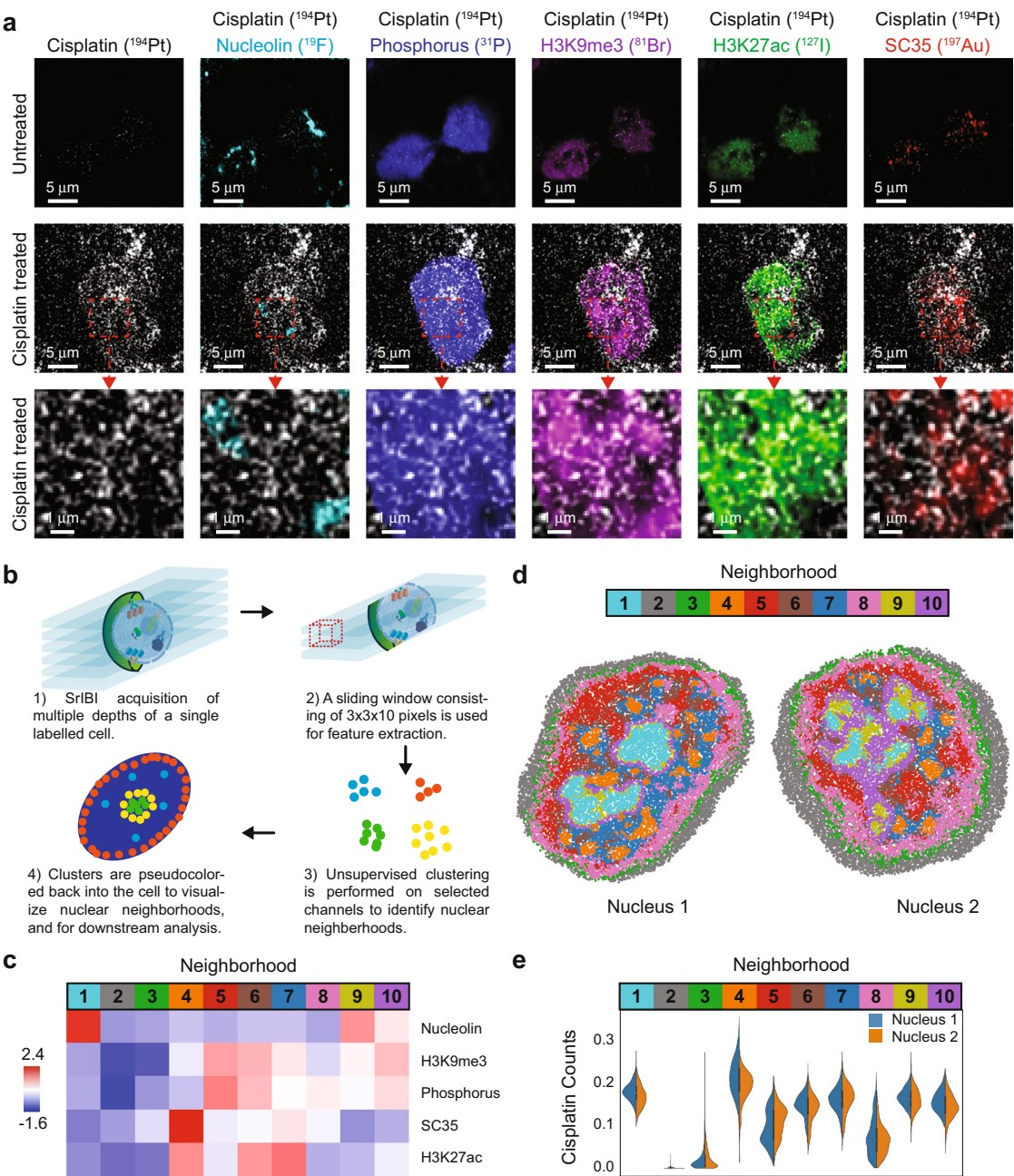

**Fig. 4 Subcellular localization of cisplatin distribution. a** Representative HD-MIBI images of TYK-nu ovarian cancer cells treated with DMSO (top row) or 15 μM cisplatin for 72 h (middle and bottom rows) and stained with anti-nucleolin-[19]F/FITC, anti-H3K9me3-[81]Br/Cy3, anti-H3K27Ac-[127]I/Cy5, and anti-SC35-biotin (detected with streptavidin-[197]Au/FITC). We simultaneously acquired images of nucleolin ([19]F), DNA ([31]P), H3K9me3 ([81]Br), H3K27Ac ([127]I), cisplatin ([194]Pt), and SC35 ([197]Au), in addition to ̄e. (First column) Cisplatin ([194]Pt) distribution within the cell. (Second to sixth columns) Overlay of ion images for cisplatin ([194]Pt; white) and a second marker. The brightness intensities for H3K9me3 and H3K27ac were lowered for cisplatin-treated images, compared to untreated images, to avoid overexposure of the cisplatin-treated images. Images consist of the sums of 100 consecutive planes. Scale bars, 5 μm. $n = 2$. **b** A schematic for the identification of nuclear neighborhoods. (1) HD-MIBI acquisition is performed on single TYK-nu cells treated with cisplatin and labeled with MoC-Abs as described in panel (**a**). (2) A sliding window of $3 \times 3 \times 10$ pixels is used for feature extraction for all channels imaged. (3) Unsupervised clustering is performed on extracted features to identify nuclear neighborhoods. (4) Identified clusters are recolored back onto the cell to visualize nuclear neighborhoods spatially. **c** A heatmap of the 10 distinctive neighborhoods identified based on the five indicated nuclear markers. The intensity of each marker is denoted by the scale bar on the left. The identified clusters of interactions, termed nuclear neighborhoods, are colored and numbered. **d** Identified nuclear neighborhoods are used to recreate the cell, resulting in distinctive structures that resemble known features in the nucleus. Two representative TYK-nu nuclei are shown. **e** Violin plots showing cisplatin counts for each nuclear neighborhood identified in panel (**c**) for the two nuclei shown in panel (**d**).

the nuclear organization diversity observed in iterative HD-MIBI images of cisplatin-treated cells using the neighborhood and spatial information alone (Fig. 5e and Fig. S26), demonstrating that the inferred neighborhoods are an appropriate representation of nuclear organization.

Since we expect the nucleus is organized with intent, certain neighborhoods should interact with each other at a higher frequency than random. We implemented a pairwise interaction permutation test for this (Fig. 5f, see "Methods" for details) and the statistical significance for each pairwise interaction is

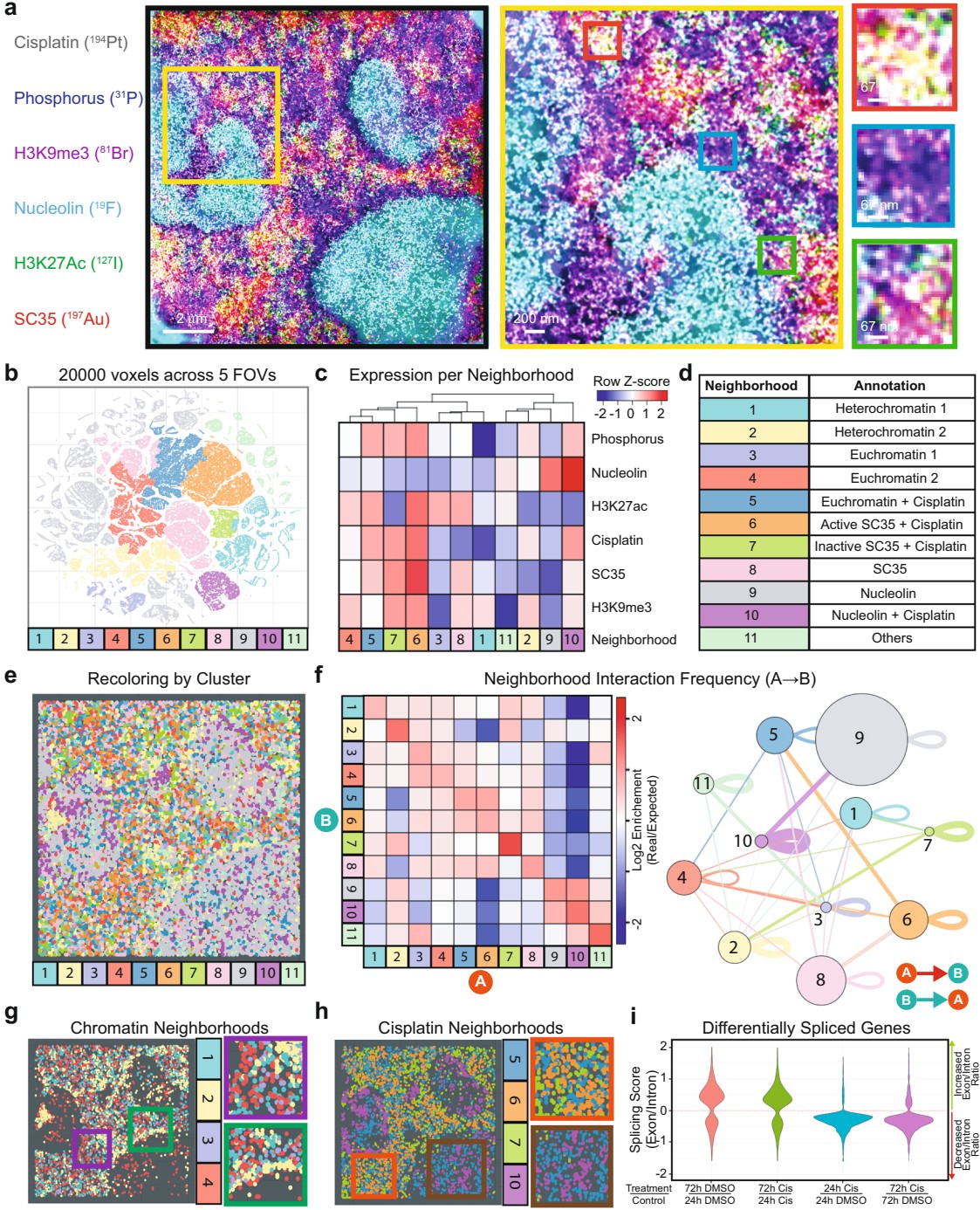

indicated (Fig. 5f and Fig. S27). As seen in Fig. 5f, there was a clear preference for co-association, or not, of certain nuclear neighborhoods.

Neighborhoods rich in nucleolin were highly self-associating (Fig. 5f, neighborhoods 9–10), indicating a clear partitioning of the nucleolin in the nucleus. Neighborhoods representing chromatin (neighborhoods 1–4) were also compartmentalized for "inactive" (neighborhoods 1 and 2) or "active" (neighborhoods 3 and 4). We observed that heterochromatin self-aggregated, was tightly packed, and surrounded the nucleolus (Fig. 5f, g, green box), whereas euchromatin was dispersed, as expected due to higher accessibility of the DNA (Fig. 5f, g, violet box). In line with previous reports we corroborated that cisplatin accumulated in the nucleolus (Fig. 5c)[45], but we further observed that cisplatin within the nucleolus spatially

segregated into two distinct neighborhoods (Fig. 5f, h, brown box, neighborhoods 5 and 10); we do not know the nature of this bilocation. In addition, extending our initial unique finding (Fig. 4f), we observed that nuclear speckle neighborhoods were distinctly enriched in cisplatin (Fig. 5c, h, orange box, neighborhoods 6 and 7; and Fig. S28).

Interestingly, the nuclear speckle neighborhoods 6 and 7 were highly self-associating but had no significant inter-neighborhood interactions (Fig. 5f, h, neighborhoods 6 and 7). Since the nuclear speckles are nuclear domains enriched in pre-mRNA splicing components, we were compelled to investigate manners in which cisplatin might interfere in cell-wide splicing of RNA. We inferred incomplete splicing (as determined by the presence of unspliced intronic sequences) from triplicate bulk poly(A) + RNA-seq

**Fig. 5 An analytical framework to identify nuclear neighborhood interactions in multiplexed super-resolved data. a** Representative HD-MIBI image of cisplatin-treated TYK-nu cells. TYK-nu ovarian cancer cells were treated with 5 μM cisplatin for 24 h and stained with anti-nucleolin-[19]F/FITC, anti-H3K9me3-[81]Br/Cy3, anti-H3K27Ac-[127]I/Cy5, and anti-SC35-biotin (detected with streptavidin-[197]Au/FITC). Images of nucleolin ([19]F), DNA ([31]P), H3K9me3 ([81]Br), H3K27Ac ([127]I), cisplatin ([194]Pt), and SC35 ([197]Au) were simultaneously acquired. A 100-μm$^2$ ROI in the nucleus was acquired by iterative HD-MIBI. Displayed are nucleolin (cyan), phosphorus (blue), H3K9me3 (magenta), H3K27Ac (green), cisplatin (gray), and SC35 (red). Denoising was performed using a k-nearest neighbor approach. An unfiltered image is shown in Fig. S24. Enlarged images from boxed regions exemplify the nuclear organization diversity. Images consist of the sums of 20 consecutive planes. **b** t-SNE map colored by the 11 identified nuclear neighborhoods. The t-SNE map is derived from 100,000 voxels of dimension $(x, y, z) = (10, 10, 5)$ pixels (20,000 voxels were randomly sampled across 5 different cells). Each point represents a voxel. Voxels grouped in distinct regions based on the expression of each marker. The 11 hierarchical clusters were identified by unsupervised hierarchically clustering, followed by manual annotation. **c** An expression profile of the mean marker expression in each of the 11 distinctive nuclear neighborhoods. The scale intensity of each marker is denoted by the color bar on the top right (Z-score, normalized to each row). **d** Annotation of each neighborhood, based on their mean marker expression profile from panel (**c**). **e** Identified nuclear neighborhoods described in panel (**d**) are used to recreate the cell, resulting in distinctive structures that resemble known features in the nucleus shown in panel (**a**). **f** Pairwise interaction frequency calculations. A permutation test was implemented to quantify the frequency of pairwise neighborhood interactions. If two points (e.g., A and B) were within 5 pixels of each other, they were defined as interacting. The mean of 1000 shuffled interaction frequencies was taken as the expected interaction frequency. (Left) The neighborhood interaction frequency map was calculated using the log2 enrichment of the real over expected number of pairwise interaction frequencies. (Right) The same interaction frequency was represented here in graph form, where size-proportional nodes represent neighborhoods, while width-proportional edges represent log2 enrichment of the real over expected interactions between neighborhoods. Only statistically significant edges (one-sided test, $p < 0.05$, see "Methods" for details) are plotted. The directionality of the edges is indicated by the color of the node it originates from. **g**, **h** Nuclear neighborhoods are colored as shown in the legend of panel (**d**) to show **g** chromatin-specific neighborhoods, and **h** cisplatin-enriched neighborhoods. Enlarged images from boxed regions exemplify different neighborhood interactions: accessible euchromatin (purple box), heterochromatin near the nucleolus (green box), an inactive/active boundary of cisplatin-containing nuclear speckles (orange box), and cisplatin-euchromatin adducts within the nucleolus (brown box). **i** Differentially spliced transcripts between control and treatment conditions were identified ($p$-adj < 0.05, from left to right, $n = 3170, 3842, 1012,$ and 1034). The splicing score was calculated, where 0 represents no change, >0 represents increased splicing and <0 represents decreased splicing in the treatment compared to control.

experiments at two time points (Fig. 5i), and calculated a splicing score (>0 indicates increased splicing, 0 indicates no change, <0 indicates decreased splicing in treatment relative to control; see "Methods" for details). RNA splicing was noticeably reduced when comparing cisplatin treatments with control at both 24 and 72 h (Fig. 5i). Thus, the reduction in spliced RNA transcripts observed by global transcriptome profiling orthogonally complemented the HD-MIBI observation that cisplatin accumulates within nuclear speckles, supporting a role for cisplatin in driving aberrant RNA maturation, and thereby potentially adding to its known roles in anti-cancer therapeutic modality.

**Distinctive cytoplasmic localization of cisplatin after multi-drug treatment.** Having established that HD-MIBI experimental and analytical workflows resolve co-localization of biological entities (neighborhoods) in cells, we next applied HD-MIBI to evaluate how resistance to drug action might arise, as exemplified by acute JQ1 and cisplatin multi-drug treatment in TYK-nu cells. JQ1 is a well-characterized bromodomain inhibitor[54] that has proven effective in a variety of cancers and tumors[55], and its action on bromodomain protein activity is known to result in extensive epigenetic architectural reorganization. Recent studies show a synergistic effect of JQ1 with cisplatin in a number of cancers, including ovarian cancer[56]. JQ1 appears to sensitize cells to platinum-based drug therapy and is considered a potential strategy for multi-drug modalities, yet the exact mechanism by which it works is not known. Further, despite the efficacy observed in vitro and in vivo, a small number of cells still survive treatment, raising the question of how cisplatin resistance develops. To address this question, the distribution of both cisplatin and its relative location to nuclear landmarks in the cell could explain its efficacy and why some cells might be resistant.

We applied HD-MIBI in cells treated for 72 h with JQ1, cisplatin, or a combination of both (Fig. S29A). Phase-contrast microscopy confirmed the reported synergistic effects of cisplatin and JQ1, as well as the presence of resistant cells that survived such treatment (Fig. S29B). We acquired 33 FOVs, each

containing 1 or 2 cells, across the four different conditions for an integrative analysis of cellular neighborhoods, identifying six functional neighborhoods present in all FOVs (Fig. 6a, b, Fig. S30–S31 and "Methods"). Strikingly, the distribution of cisplatin in single or multi-treated cells (Fig. 6c and Fig. S32) was excluded from the nucleus of cells resistant to treatment with both JQ1 and cisplatin (Fig. 6c, bottom).

We applied our neighborhood interaction analysis to discern changes to the cellular architectural due to the different treatment conditions (Fig. 6d and Fig. S33). While the majority of cellular organization appeared to be largely conserved, we observed that interactions of euchromatin with nucleolin (neighborhoods 3 to 1), as well as euchromatin with heterochromatin (neighborhoods 3 to 2), were enriched in the multi-drug treated cells (Fig. 6d and Fig. S33). In addition, the proportions of nucleolin and chromatin modifications were also affected by the combinatory treatment (Fig. 6d). These results suggest a specific but not global cellular reorganizational response for resistance to multi-drug treatment.

We confirmed that cisplatin positive voxels were negligible in DMSO and JQ1 treatments and observed a minor decrease in the combined treatment compared to cisplatin alone (Fig. S34A). We then determined whether cisplatin was more subtly redistributed into different cellular neighborhoods by the action of JQ1 (as seen visually in Fig. 6c). The fraction of cisplatin positive voxels within each neighborhood, in the presence and absence of JQ1, confirmed a significant reduction of cisplatin within the nuclear, but not cytoplasmic neighborhoods (Fig. 6e and Fig. S34B). Thus, our analytical pipeline quantitatively confirmed the observation: cisplatin is excluded from the nucleus of cells that survive JQ1 and cisplatin co-treatment. These results support a model wherein selective depletion of cisplatin from the nucleus in cells enables them to survive combination platinum-based therapy and bromo-domain inhibition.

**Discussion**

HD-MIBI is a mass spectrometry-based method enabled by a new set of isotope-labeled antibody reagents, MoC-Abs, that

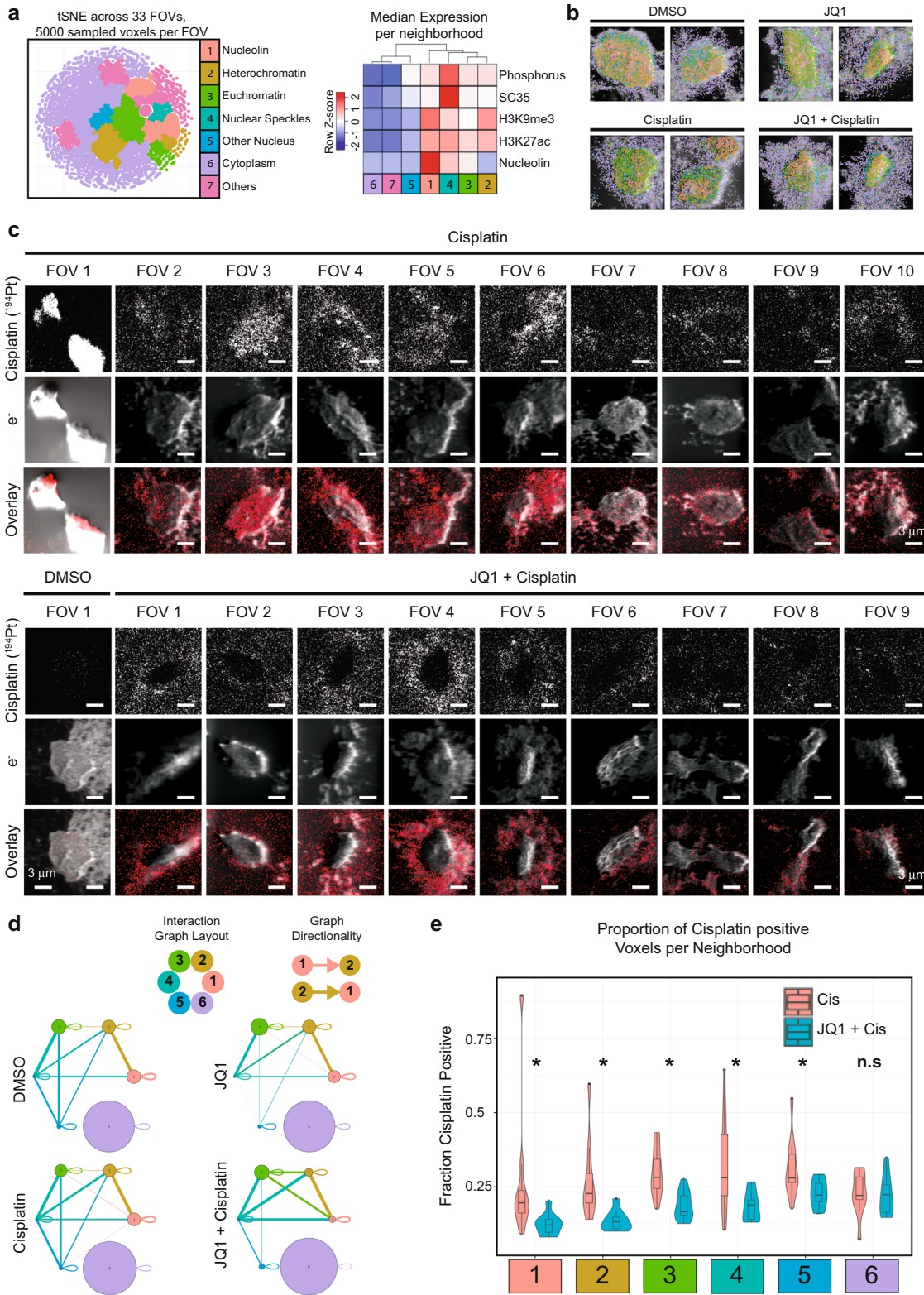

allow for the specific detection of proteins using MIBI coupled to a cesium primary ion beam. This method builds upon prior biological applications using nanoSIMS to enable targeted high-definition, 3D, multiplexed imaging of different biomolecule types in single cells[17,25]. HD-MIBI detection of MoC-Abs yielded results akin to those obtained using standard confocal light microscopy but use of iterative HD-MIBI imaging resulted in lateral resolutions of ~30 nm. Application of unsupervised

dimensional reduction methods enabled the identification of subcellular structures and localization of the drug cisplatin. As most drugs and other small molecules (lipids, metabolites, etc.) can be isotopically labeled with isotopes such as $^{13}$C or $^{15}$N, HD-MIBI will allow for study of localization of a variety of agents. Drug metabolism in and out of cells might also be determined by such imaging. Understanding the subcellular localization of drugs may maximize the success rates of drug

**Fig. 6 HD-MIBI based identification of drug relocalization in resistant cells upon multi-drug treatment. a** 5000 sampled voxels from 33 FOVs were combined for analysis. (Left) Dimensional reduction with t-SNE, followed by hierarchical clustering, identified seven distinct cellular neighborhoods (left panel). (Middle) These neighborhoods were then annotated based on their median expression profiles (upper right panel), as well as visual confirmation. $n = 8$ cells for DMSO, $n = 6$ cells for JQ1, $n = 10$ cells for cisplatin, and $n = 9$ cells for JQ1 + cisplatin examined in 1 experiment. **b** The secondary electron image for each cell is overlaid with the locations of identified neighborhoods. Two representative FOVs are shown for each treatment. Scale bars, 3 μm. **c** The raw secondary electron and cisplatin images for all FOVs treated with cisplatin or JQ1 + cisplatin, together with a representative FOV from the DMSO control, are shown. Scale bars, 3 μm. **d** Neighborhood interaction frequencies between each neighborhood represented here in graph form. Size-proportional nodes represent neighborhoods, while width-proportional edges represent log2 enrichment of the real over expected interactions between neighborhoods. Only statistically significant edges (one-sided, $p < 0.05$, see "Methods" for details) are plotted. The directionality of the edges is indicated by the color of the node it originates from. **e** Violin and box plots showing the proportion of cisplatin positive voxels for each neighborhood, comparing cisplatin to JQ1 + cisplatin conditions (* indicates $p < 0.05$ for a two-sided Wilcoxon test; n.s. not significant). The center of the box corresponds to the median. The minima and maxima bound of box correspond to the 25th and 75th percentiles, respectively. The whiskers extend from the minima and maxima bounds of box to the largest value no further than 1.5 times the inter-quartile range. The outliers are shown as dots. See Figure S34B for a representation of the variability on each single cell. Cisplatin (Cis). $n = 8$ cells for DMSO, $n = 6$ cells for JQ1, $n = 10$ cells for cisplatin, and $n = 9$ cells for JQ1 + cisplatin examined in 1 experiment.

candidates in clinical trials, holding promise for a more efficient and cost-effective drug discovery process[57].

With the advancement of methods to probe subcellular contents at high resolutions, we introduce here analytical frameworks that allow biologists to make sense of the data. Combining HD-MIBI and these analyses, we identified nuclear neighborhoods within single cells: spatial locations within the nucleus with similar molecular components and possibly, functions. This allowed the observation of cisplatin enrichment within nuclear speckles, nuclear domains enriched in pre-mRNA splicing components. Complemented with a decreased mRNA splicing on cisplatin-treated cells as showed by our global transcriptome analysis and work from others[58], our data support a functional role for cisplatin in affecting mRNA maturation, possibly through localization to the periphery of nuclear speckles or cross-linking to certain RNAs or pre-spliceosomes in a manner that precludes appropriate splicing. Cisplatin within the nucleolus was spatially segregated into a distinct neighborhood. Since the nucleolus is itself segregated into three major components, understanding of cisplatin location within that context might provide unique detail on preferential mechanisms of action in situ.

The application of HD-MIBI to dissect mechanisms of cisplatin resistance under sensitization by JQ1 yielded the striking result that surviving cells were nearly completely excluded cisplatin from the nucleus. This implicates a nuclear pore-based pump that functions in multi-drug resistance. Interestingly, multi-drug resistance to cisplatin and other drugs with surface receptors is a well-documented phenomenon[59] and begs the question of whether other classes of small molecule pumps have evolved to maintain the nucleus free from certain metabolites or environmental poisons, mechanisms that might act in concert for a more efficient pumping out of drugs from the cell. Beyond that, this study sets the stage for future work studying the action of drugs with no need to be structurally altered, by harnessing intrinsic elements or labeling during drug synthesis with alternative stable isotopes ($^{13}$C, $^{15}$N, etc.).

Although HD-MIBI has theoretical resolutions depending on tag size (for instance an antibody might be larger than the target), it is currently hindered by low ion yields at high resolutions, the number of isotopes available for labeling, the number of available detectors for higher multiplexing in a nanoSIMS, and potential distortions through the etching process. Improvements to the ion yield can be linearly ameliorated through the attachment of more labeling isotopes per probe (which may result in slight loss of resolution). The MoC-Abs used in this work each contain 48–72 isotopic labels. An increase in labels per antibody can be achieved by increasing the number of substituted nucleotides or increasing the length of the labeling oligonucleotide. Results demonstrated that a twofold

increase in oligonucleotide length together with a ~fourfold increase in labeled nucleotides enable high-resolution imaging using HD-MIBI (Fig. S35) but further increases would require additional validation to ensure localization estimations can be maintained (compact DNA origami techniques might be useful here[60]). Such an updated MoC-Ab design containing 168–252 isotopic labels will theoretically allow detection limited down to a single antibody-targeted protein (Supplementary Note 1). The use of nucleotide-based amplification methods[35,61–63] could also be applied for certain low copy number targets. Better estimations of the placement of the epitope at which the antibody interacts (through use of centroid estimations) might be obtained by directional labeling of the antibody: If one isotope was located at the 5′ end of the mass-oligonucleotide and another isotope at the 3′ end, it would provide a "pointer" to the epitope is located. The atomic elements used for HD-MIBI can be expanded by synthesizing isotope-derivatized nucleotides with previously reported protocols[64,65]. Moreover, a recently developed cesium ion beam has attained spot sizes as small as 2 nm, which promises to enable more precise localizations[66]. Physical expansion of biological samples, mediated by polymer swelling as previously described[67], would further increase the resolution of HD-MIBI. Other primary beam sources, such as helium- or neon-based beams, which have beam sizes of 0.5 and 2 nm, respectively, have been recently adapted for SIMS[68]. Successful implementation of these technologies for effective targeted imaging, through methods akin to HD-MIBI, may require additional signal amplification and chemistry modifications to achieve more efficient ionization efficiencies. These improvements may enable 3D imaging of single antibodies, hence opening the door for such opportunities as isotope barcoding to increase the number of simultaneously measured channels. For example, with the seven-detector design presented here, a triple-barcoded labeling would theoretically enable deconvolution of 35 targets (Supplementary Note 2). If negative secondary ion detection is not required, the new Hyperion II ion source installed in some NanoSIMS provides lateral resolution down to 40 nm[69], and it is compatible with lanthanide-conjugated antibodies.

Analysis of samples with uneven topography and presenting locations with heterogeneous atomic density, such as the cells shown in this work, results in potential for distortions and differential sputtering yields through the etching process that affect 3D reconstruction. Atomic force microscopy (AFM) is a powerful method to correct for such distortions by providing a snapshot of the sample topography before and after the analysis[70,71]. However, artefacts arising from sample transfer and distinct chamber environments are still a limitation. In situ AFM, the integration of AFM with SIMS instruments, would be the ideal theoretical solution but it is currently hindered by instrumental availability

and technical challenges[72]. Instrument improvements, reconstruction of serially sectioned cells, or photogrammetric 3D reconstruction[73], also hold promise for future exploration. Cisplatin has been observed to form adducts with DNA and RNA, while forming cross-links to proteins. After sample processing, unbound cisplatin is washed away but covalently linked cisplatin remains. Cryosectioning together with genetically-encoded chemical tags[27,74], can provide a workflow for comparative studies into unbound versus bound intracellular drug distribution.

Growing evidence implies that interactions among different cell subtypes in tissues are critical for defining tissue morphology and function[3,14]. Administering isotopically labeled drugs in vivo[23] and subsequent tissue staining with MoC-Abs will allow a better understanding of the bio-distribution of lipids, small drugs, or metabolites. However, while HD-MIBI is not especially well-suited for whole tissue imaging due to its limited FOV, an exciting alternative would be to combine HD-MIBI in a correlative fashion with highly multiplexed tissue imaging techniques, such as the MIBI[75]. HD-MIBI re-scan of a region of interest identified using those techniques would allow studies aimed to understanding the finer details of membrane contacts or epigenetic changes in relevant cell-to-cell interactions occurring in vivo.

Super-resolution light microscopy of multiple biomolecules simultaneously on the same sample is challenging. While recent applications overcome this issue by using cycling of oligonucleotide probes[2,3,5,8], these methods suffer from drawbacks such as sample drift between cycles, low axial resolution, photobleaching, and long acquisition times. Current methods such as the tetrapod PSF and DNA-PAINT are promising solutions to some of these limitations[76–78]. Since HD-MIBI acquires data on multiple biomolecules simultaneously, it can serve as an alternative to certain super-resolution light microscopy techniques. Samples can be stored for extended periods as the isotopes detected are stable—which might not be the case for many fluorophores used in photon-based imaging. HD-MIBI takes advantage of reagents that can be found in standard molecular biology laboratories but requires specialized equipment for data acquisition (Supplementary Note 3). Being an antibody-based technique, HD-MIBI inherits the advantages and disadvantages associated with such reagents. A vast amount of literature on chemical modifications of antibody and oligonucleotides readily supports further MoC-Ab adaptations based on individual needs. In addition, the conjugation and staining processes detailed here are readily compatible with carrier-free commercially available primary antibodies, which will likely facilitate HD-MIBI accommodation to the scientific community. Extension of fluorescent-based methods for specific nucleic acid staining, like CRISPR/Cas-FISH[79], Oligopaints[34,80], and ATAC-see[81], to HD-MIBI will provide a unique opportunity for an integrative understanding of cellular processes at the nanoscale beyond current means. However, incomplete epitope saturation and steric hindrance are potential limitations associated with HD-MIBI, especially for crowded environments like the nucleus. At the highest HD-MIBI resolutions, the use of antibodies is linked to an increase of uncertainty while inferring actual protein position from the detected signal due to their size (~150 kDa), a caveat shared with super-resolution light microscopy methods[82]. Translation of smaller probes for specific protein detection to HD-MIBI, such as isotopically labeled nanobodies[83] and SOMAmers[43] have the potential to enhance location precision.

Multiplexed imaging of an entire cell, at resolutions enabled by HD-MIBI, introduces new data processing challenges in how to interpret the results. To this end, we designed and implemented an analytical framework that leverages on established methods to infer biological meaning. For example, hierarchical clustering and t-SNE are able to simplify multiplexed HD-MIBI data across multiple cells. Combined with neighborhood interaction maps, we are able to reveal interactions within the subcellular domain not obvious to the naked eye. Future improvements should include better methods to extract data meaningfully, including their combination with conventional analysis. The ability to learn and infer information about other unlabeled but related biomolecules within the cell will provide a way to describe an entire cell with a limited number of parameters[84,85]. Further integrative analysis, particularly incorporation of biophysical and genomic information, will also allow more accurate modeling of nuclear structures[8,86,87].

HD-MIBI is an imaging method that can be applied in single-cell studies of the diverse molecular interactions in subcellular microenvironments. It has been recently reported that antineoplastic drugs, including cisplatin, concentrate in specific nuclear protein condensates in vitro[88]. Combination of these in vitro assays with HD-MIBI, together with unbiased or population-based methods, such as small molecule screens and multiplexed proteomics, promises to reveal new understanding into metabolic pathways or mechanisms of drug resistance. Here we applied HD-MIBI imaging to cisplatin small molecule distribution in cells, and in the process discovered previously unappreciated mechanisms by which cisplatin either functionally acts or by which cells might become resistant to its modes of action. This has the potential to drive new therapeutic discoveries and inform decision-making processes of such small molecules wherein visualization allows more direct interpretation of mechanism and function at the molecular scale.

## Methods

**Cell culture.** TYK-nu cells (a gift from Dr. Fantl at Stanford) were grown in Eagle's minimal essential medium (American Tissue Culture Collection). HeLa cells (ATCC) were grown in DMEM (Gibco, Invitrogen). Jurkat cells (Clone E6.1; ATCC) were grown in RPMI (Gibco, Invitrogen). Media were supplemented with 10% heat-inactivated fetal bovine serum, 100 U/mL penicillin (Gibco, Invitrogen), and 100 mg/mL streptomycin (Gibco, Invitrogen). Cells were cultured in a humidified cell incubator at 37 °C with 5% $CO_2$ conditions and split with TrypLE Express (Gibco, Invitrogen) every 2–3 days.

**MoC-Ab preparation.** Oligonucleotides (Table S1) were synthesized at the Stanford Protein and Nucleic Acid Facility with internal isotope-derivatized nucleotides, a fluorophore at the 3′ position, and a maleimide cycloadduct at the 5′ position. The maleimide was deprotected by a retro Diels-Alder reaction. The lyophilized oligonucleotide was suspended in 1 mL anhydrous toluene (MTX07327, Millipore) for 4 h at 90 °C, washed four times with anhydrous ethanol, and solubilized in buffer C (2 mM Tris, 150 mM NaCl, 1 mM EDTA, pH 7.2). The oligonucleotide concentration was determined using a Nanodrop spectrophotometer. Aliquots of 8.5 nmol were prepared, lyophilized overnight, and stored in a desiccator at −20 °C. Antibodies (Table S2) in carrier-free PBS were conjugated to the deprotected oligonucleotides. Briefly, 50 μg of antibody was loaded into a 50-kDa 0.5-mL centrifugal filter column with 400 μL PBS and reduced with 400 μL reduction buffer (PBS with 2.5 mM TCEP and 2.5 mM EDTA) for 30 min at room temperature. Antibodies were then washed with 400 μL of C buffer into a 50-kDa 0.5-mL centrifugal filter column and conjugated to 8.5 nmol of oligonucleotide in 400 μL conjugation buffer (buffer C with 0.5 M NaCl) for 2 h at room temperature. Antibodies were washed five times with 400 μL of high-salt PBS (PBS with 1 M NaCl), diluted into storage buffer (Candor PBS Antibody Stabilization Solution with 0.5 M NaCl and 5 mM EDTA), and stored at 4 °C. Each MoC-Ab was titrated by immunofluorescence using HeLa cells, as exemplified in Fig. S3, and the staining pattern was compared to the staining pattern of the unconjugated antibody.

**Lanthanide-conjugated and biotin-conjugated antibodies.** Antibodies (Table S2) in carrier-free PBS were conjugated to metal-chelated polymers (MaxPAR Antibody Conjugation Kit, Fluidigm) or Sulfo-NHS-SS-Biotin (A39258, ThermoFisher) according to the manufacturer's protocol. Antibodies were diluted to 0.2 mg/mL in Candor PBS Antibody Stabilization Solution and stored at 4 °C.

**Evaluation of cisplatin accumulation in cells by mass cytometry.** TYK-nu cells were cultured in 0.5, 5, or 50 μM of cisplatin (P4394, Sigma-Aldrich) for 24 h, washed, and treated with 1 μM Rh-intercalator (201103B, Fluidigm) for 15 min to

discriminate dead from live cells. Cells were analyzed in a CyTOF2 instrument (Fluidigm).

**Evaluation of cisplatin accumulation in cells by super-resolution ion beam imaging**. 50,000 TYK-nu cells were seeded into each well of a 24-well plate containing a 7 mm × 7 mm silicon wafer. The next day, DMSO control or 15 μM cisplatin (201194, Fluidigm) was added. Intracellular staining was performed 24 h later as described below.

**Cisplatin and JQ1 treatments**. 50,000 TYK-nu cells were seeded per well of a 24-well plate with 7 mm × 7 mm silicon wafers. The next day, DMSO control (equivalent volume as cisplatin), 15 μM cisplatin (201194, Fluidigm), 250 nM JQ1 (500586-1SET, Sigma-Aldrich) or a combination of cisplatin and JQ1 were added. Intracellular staining was performed 72 h later as described below.

**Intracellular staining**. Compositions of buffers used for staining are given in Table S3. Cells were fixed and permeabilized for 30 min at 4 °C in Fixation/ Permeabilization buffer. Cells were then gently washed three times with Wash Buffer and blocked in Block Buffer 1 for 30 min at room temperature, washed three times with Wash Buffer, blocked in Block Buffer 2 for 30 min at room temperature, and washed three times with Wash Buffer. The cells were stained with a mixture of MoC-Abs in Reaction Buffer for 3 h at room temperature. Following staining with MoC-Abs, the Reaction Buffer was removed by gently touching the sample with a precision wipe (05511, Kimtech Science). Cells were then washed twice in Wash Buffer and incubated with Reaction Buffer containing FluoroNanogold Streptavidin (1:40 titer, 7016, Nanoprobes) for 30 min at room temperature. Cells were washed twice with Wash Buffer before preparation for confocal microscopy or HD-MIBI analysis. Cell fixation was performed in 1X PBS with 1.6% paraformaldehyde for 10 min at room temperature, followed by permeabilization in methanol (10 min at −20 °C) or with Triton X-100 (0.5% in 1X PBS, 10 min at room temperature). The staining produced comparable results between the two permeabilization methods for most of the antibodies tested.

**Tissue staining**. Tissue sections of 4-μm thickness from a fresh-frozen adult human skin OCT block were prepared on a cryostat and stored at −80 °C. Tissue sections were placed from the −80 °C into a chamber containing Drierite (230001, Cole-Parmer) for 2 min at room temperature. Tissue sections were then incubated in acetone for 10 min at room temperature and then placed again into a chamber containing Drierite for 2 min. Tissue sections were hydrated in 1X PBS containing 0.5% BSA and 5 mM EDTA for 2 min at room temperature before fixation in 1X PBS containing 0.5% BSA, 5 mM EDTA, and 1.6% PFA for 10 min at room temperature. Tissue sections were washed twice in 1X PBS containing 0.5% BSA and 5 mM EDTA and then incubated in 1X PBS containing 0.5% BSA, 5 mM EDTA, and 1 M NaCl for 10 min at room temperature. Tissue sections were blocked with 1X PBS containing 0.5% BSA, 5 mM EDTA, 0.5 M NaCl, 200 μg/mL Sheared Salmon Sperm DNA, and 10 μg/mL Human FcX Block (422302, Biolegend) for 30 min at room temperature and then stained in blocking buffer containing anti-H3K9me3 MoC-Ab (1:50 dilution). Tissue sections were then washed twice in 1X PBS containing 0.5% BSA and 5 mM EDTA before preparation for confocal microscopy or HD-MIBI analysis. The frozen skin OCT block was collected under a protocol approved by the Institutional Review Board at Stanford University (protocol no. 35324). Individuals donating fresh surgical tissue provided informed consent.

**Confocal microscopy**. For confocal microscopy analysis, cells were grown in 12-mm diameter glass coverslips (72226-01, Electron Microscopy Sciences) until 80–90% confluent and stained as described. After staining, cells were washed three times with 1X PBS, rinsed once with water, and mounted using Vectashield with DAPI (H-1200, Vector Laboratories). Secondary antibodies anti-mouse-Alexa488 (1:2000 titer, 4408S, Molecular Probes) or anti-mouse-Alexa647 (1:2000 titer, 4410S, Molecular Probes) were used to detect MoC-Abs or unconjugated antibodies. Cells were analyzed on a LSM 880 (Zeiss) with a ×63 oil-immersion objective (Zeiss Plan-Apochromat 63x/1.4 Oil).

**Super-resolution ion beam imaging**. For HD-MIBI analysis, cells were grown on silicon wafers (7 mm × 7 mm or 18 mm × 18 mm, Silicon Valley Microelectronics). Wafers were rinsed twice in methanol, air-dried with compressed air, washed with ethanol for 10 min, and rinsed three times with sterile 1X PBS in a cell culture hood prior to cell seeding. When cells reached 80–90% confluence, they were stained as described above. After staining, cells were washed twice in 1X PBS, fixed for 5 min in 1X PBS containing 2% glutaraldehyde (Post-Fixation buffer), and rinsed five times with water. Cells were dehydrated using a graded ethanol series, air-dried in a desiccator chamber, and stored at room temperature in a vacuum desiccator until analysis. HD-MIBI images were acquired with the NanoSIMS 50 L mass spectrometer (Cameca) at Stanford University using the CAMECA Microbeam Cesium Source. Before each experiment, tuning of the primary optics, secondary optics, and mass spectrometer was performed. The detectors were tuned by identifying each isotope of interest in an air-dried drop

containing all isotopes and placed on a silicon wafer. The ion detectors were set as follows: detector 1, $^{12}C$; detector 2, $^{19}F$; detector 3, $^{31}P$; detector 4, $^{81}Br$; detector 5, $^{127}I$; detector 6, $^{194}Pt$; and detector 7, $^{197}Au$. Secondary electrons were detected in parallel with ion information. The secondary ion peaks and E0S were tuned before the acquisition of each new cell to correct for magnetic drift. All images were collected with a 1-ms dwell time per pixel. The raster size, number of pixels, repeat scans over the same area, and total scan time for each image are listed in Table S4. Images were opened using the ImageJ plugin openMIMS developed at the Brigham and Women's Hospital Center for NanoImaging (github.com/BWHCNI/OpenMIMS). For each HD-MIBI image shown in this manuscript, individual planes described in Table S4 were summed and gaussian blurred with a sigma of one using ImageJ/FIJI (NIH). Three-dimensional rendering was performed using Imaris (Oxford Instruments), and figures prepared using Adobe Illustrator.

**Resin embedding and silicon quantification**. Jurkat cells were fixed overnight in post-fixation buffer (Table S3), washed three times in PBS, processed with a standard epoxy embedding protocol (Stanford Cell Sciences Imaging Facility), and the resulting cell pellet sectioned to a 100-nm thickness on silicon wafers (7 mm × 7 mm). In nanoSIMS, regions on the edge of the organic material were identified and images were acquired at different primary beam currents. The silicon signal ratio between the organic material and the substrate was then calculated for each depth in three independent regions of the image.

**Nuclear neighborhood analysis**. A mask for each nucleus was created using the phosphorus channel, and 20,000 3D voxels, each of dimension $(x, y, z) = (3, 3, 10)$ pixels, were extracted from each of two cells by random sampling from the middle 40 Z-planes (for a total of 40,000 voxels). Each channel was z-normalized across all pixels contained within the sampled image regions. Unsupervised hierarchical clustering was performed using the markers SC35, H3K9me3, phosphorus, nucleolin, and H3K27Ac to identify 10 clusters across the 20,000 extracted voxels. The average expression of each marker from both cells was then plotted in a heatmap form. Each voxel was colored by its cluster, and replotted on a X–Y plane to recreate the cell. The count distribution of cisplatin within regions assigned to each cluster was computed for each cell.

**Iterative HD-MIBI data analysis**. The low pixel signal intensity values of iterative HD-MIBI due to the much lower current during data acquisition than in conventional HD-MIBI is a barrier for robust data analysis. We adapted a filtering strategy described in Keren et al.[14] to filter noise from signal using a k-nearest neighbor (KNN) approach. The assumption for this approach is that in sparse data, the signal density offers more information than signal intensity. Thus, regions with dense signals are more likely to contain true signals, compared with regions with sparse signals. We first summed the image data (1024 × 1024 pixel) across 20 Z-planes, and then measured the average distance between positive pixels for the nearest 25 neighbors within each channel. We next applied a cutoff to distinguish between signal (regions of high signal density) and noise (regions with sparse signal density), to create a mask for each channel. The masks were applied to each individual Z-plane to create a denoised image for subsequent analysis. The 100 pixels around each border were discarded to avoid potential edge effects. A 3D sliding window of dimension $(x, y, z) = (10, 10, 5)$ pixels and step-size of $(x, y, z) = (5, 5, 3)$ was applied to the image to calculate the mean of each channel within the voxel. We randomly sampled 20,000 voxels across each image (from 5 different cells) to obtain a final 100,000 voxels. Counts were log2 transformed after adding a "0.0001" value to avoid zeros. Unsupervised hierarchical clustering was performed on all voxels using the markers SC35, H3K9me3, phosphorus, nucleolin, H3K27Ac, and cisplatin. A t-distributed stochastic neighbor embedding (t-SNE) using a Barnes-Hut implementation was performed, and each identified cluster was differentially colored. The means of each cluster was also plotted and subsequently manually merged by similarity into 11 final clusters. The t-SNE plot was also colored by the cell of origin of each voxel to ensure minimal batch effects.

**Nuclear neighborhood interaction frequency**. To identify and test for the significance of how frequently clusters interacted (as defined by the number of pixels separating voxels from each other), we implemented a permutation test. First, the Euclidian distance in 3D space between voxels was calculated (Eq. 1).

$$Dist = \sqrt{(x_1 - x_0)^2 + (y_1 - y_0)^2 + (z_1 - z_0)^2} \qquad (1)$$

Next, we defined interacting voxels as those less than or equal to 5 pixels away from each other (between the center of each voxel). After 1000 permutations, where the cluster annotation was shuffled, and the p-value calculated on either tail (Eq. 2).

$$P\ value = \frac{(\sum Permutations > (<) Real\ data) + 1}{\#Permutations + 1} \qquad (2)$$

Essentially, 1 plus the number of permutations with distances smaller than (or greater than) the real data was divided by 1 plus the total number of permutations

performed. Finally, the fold enrichment of interactions for real data over the mean interactions from the permutation tests was calculated (Eq. 3).

$$Log2\ Fold\ enrichment = log2\ \frac{Real\ data}{mean(\#Permutations)} \qquad (3)$$

*Cisplatin treatment, RNA isolation, and sequencing library preparation.* Triplicates of TYK-nu cells were grown as described above, with 15 μM cisplatin or DMSO control added for 24 or 72 h.

RNA was extracted using the PureLink RNA Mini Kit (ThermoFisher, USA), following the manufacturer protocol for monolayer cells. The NEBNext Poly(A) mRNA magnetic Isolation Module (E7490, New England Biolabs, USA) was used for mRNA isolation and the sequencing library prepared using the NEBNext Ultra II Directional RNA Library Prep Kit for Illumina (E7760) and NEBNext Multiplex Oligos for Illumina (E7500). Library quality was verified on a Bioanalyzer 2100 (Agilent Technologies, USA) and the library sequenced on a single lane of Hi-Seq 4000 (2 × 100 bp) at the Stanford Center for Genomics and Personalized Medicine sequencing Center.

**RNA-seq data processing.** Sequencing library quality was checked using FASTQC v0.11.8 at the default settings. Trim Galore v0.6.0 was used to trim out adapter and poor-quality sequences at default settings. Reads were aligned against the human genome (GRCh38 build), using HISAT2 v2.1.0. Gene annotation was retrieved from Ensembl (Homo_sapiens.GRCh38.98.chr.gtf).

Exon/Intron fragment counting was performed as described (https://github.com/csglab/CRIES). In short, individual GTF files were constructed for Exons and Introns. Subsequently, HTSeq-count was used to count the fragments mapping to Exons or Introns for each individual gene. A '-m intersection-strict' flag was used during Exon counting, and a '-m union' flag was used during Intron counting.

**Immature enrichment score analysis.** A logistic regression model was applied using the R glm (family = binomial) function to identify genes in which the Exon/Intron ratio was significantly affected by the treatment condition. Significant genes were selected if the Benjamini & Hochberg adjusted *p*-value < 0.05. Since the exon/intron ratio was highly dependent on individual genes, we devised a splicing score to facilitate visual representation of the data (Eq. 4).

$$IE_{geneA} = log2\ \frac{\sum_{Treatment}^{Replicate}\frac{Count_{ExonA}}{Count_{IntronA}}}{\sum_{Control}^{Replicate}\frac{Count_{ExonA}}{Count_{IntronA}}} \qquad (4)$$

**Analysis of resistance arising in multi-drug treatment.** A 3D sliding window of dimension $(x, y, z) = (3, 3, 7)$ pixels and step-size of $(x, y, z) = (2, 2, 5)$ was applied to the middle 40 Z-planes of each FOV to calculate the mean of each channel within the voxel. The 12 pixels around the border of each FOV were discarded. We then scaled the mean voxel counts from each channel (with the exception of cisplatin) per FOV to 1 and then applied a 20–95 percentile normalization. Since not all cells were treated with cisplatin, this normalization would skew non-treated cells with artificially high cisplatin levels. Thus, cisplatin was normalization globally with a lower limit of 0.01 and an upper limit of 0.05, before scaling to 1. Voxels from each FOV in the bottom 20% based on $^{12}C$ counts were filtered out, to remove voxels not associated with organic material (aka empty slides). This resulted in ~62,000 final voxels for each FOV. Exactly 5000 voxels were sampled from each FOV, and all channels except for $^{12}C$ and $^{194}Pt$ were used for dimensional reduction with t-SNE. Hierarchical clustering with the same channels used for t-SNE was performed using the Euclidean distance on the 2D t-SNE plot to segregate 40 clusters, which were then annotated and merged according to their expression profile for each channel as well as spatial positions. Seven final neighborhoods were identified, and neighborhood 7 excluded from further analysis as its spatial distribution was inconsistent across FOVs, and undeterminable whether it was nuclear or cytoplasmic. Cummings plots for estimation statistics were generated using the dabestr package (https://github.com/ACCLAB/dabestr).

**Statistics and reproducibility.** All experiments have been repeated independently three or more times with similar results, unless specified in the figure legend.

**Reporting summary.** Further information on research design is available in the Nature Research Reporting Summary linked to this article.

## Data availability

All data is available in the main text or the supplementary materials. Relevant raw data generated and analyzed during the current study are available from the corresponding author on reasonable request. All sequencing data are deposited to GEO under Accession Number GSE141138. The human genome (GRCh38.98) is from Ensembl [http://ftp.ensembl.org/pub/release-98/gtf/homo_sapiens/]. The alignment index for HISAT2 is [https://genome-idx.s3.amazonaws.com/hisat/grch38_genome.tar.gz]. The data analyzed with new code has been made publicly available at https://github.com/djstar/HD-MIBI.

## Code availability

New code accompanying this manuscript is available at https://github.com/djstar/HD-MIBI.

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

## Acknowledgements

We thank Drs. Michael Angelo, Maria Angulo-Ibanez, and Maurice Lee for critical discussions and reading the manuscript. We thank Angelica Trejo and Gina Jager for technical contributions. We are grateful to the Protein and Nucleic Acid Facility of Stanford University for production of oligonucleotides and technical support. X.R.-C. is supported by a long-term EMBO fellowship (ALTF 300-2017). S.J. is supported by a Stanford Dean's Fellowship and the Leukemia & Lymphoma Society Career Development Program. A.F.C. holds a Career Award at the Scientific Interface from Burroughs

Wellcome Fund and a National Institutes of Health K25 Career Development Award (K25AI140783). F.A.B. holds a Human Frontier Science Program long-term postdoctoral fellowship. This work was supported by the National Institutes of Health (NIH) 5R01NS08953304, 5U54CA14914505, 5UH2AR06767603, 5R25CA18099304, 5R01GM10983604, 5R01CA18496804, 5R01GM10983604, Department of the Army W81XWH-12-1-0591 and W81XWH-14-1-0180, Bill & Melinda Gates Foundation OPP1113682, and the Rachford and Carlota A. Harris Endowed Professorship to G.P.N. Part of this work was performed at the Stanford Nano Shared Facilities (SNSF), supported by the National Science Foundation under award ECCS-1542152. This work also used the Genome Sequencing Service Center by Stanford Center for Genomics and Personalized Medicine Sequencing Center, supported by the grant award NIH S10OD020141, and the Diabetes Genomics and Analysis Core of the Stanford Diabetes Research Center supported by grant award NIH/NIDDK P30DK116074.

## Author contributions

X.R.-C. designed and performed most of the experiments, analyzed and interpreted the data, and wrote the manuscript. S.J. designed and performed experiments, implemented and performed most of the analysis, analyzed and interpreted the data, and wrote the manuscript. Y.B. and B. Z. performed experiments. G.B. and S.B. designed the initial nuclear neighborhood analysis, analyzed and interpreted data. A.F.C. and G.H. assisted in experimental design. C.-M.K.H. performed experiments and assisted in optimizing protocols used in MoC-Ab staining. C.H. assisted in data acquisition and experimental design. S.-Y.C. designed and performed experiments, analyzed and interpreted data, and wrote the manuscript. F.A.B. assisted in experimental design. G.P.N. was responsible for funding acquisition, assisted in experimental design, interpreted data, and wrote the manuscript.

## Competing interests

The authors declare no competing interests.
