## [Peer Review File · Nature Communications]

Reviewers' Comments:

Reviewer #2:

Remarks to the Author:

In this manuscript, Rovira-Clave and colleagues report the development of super-resolution ion beam imaging (srIBI), a mass spectrometry-based imaging technique capable of high-parameter imaging in 3D. The authors describe in details the design and validation of mass-oligonucleotide-conjugated antibodies (MoC-Ab), which they use to detect and image specific proteins using srIBI. They also validated multiparametric imaging and the identification of subnuclear structures. The manuscript also describes the application of srIBI for 3D cellular imaging and reconstruction at nanoscale level.

The authors apply multiparametric srIBI to image cisplatin distribution in TYK-nu cells. They report an enrichment of cisplatin in nuclear speckles, which are sites of splicing factor accumulation, suggesting a link between cisplatin and mRNA splicing. RNA-seq analysis further suggest an increased intron retention in cells treated with cisplatin. Finally, srIBI imaging of cells resistant to cisplatin and the BET inhibitor JQ1 reveals that cisplatin is excluded from the nucleus of these cells, suggesting that nuclear exclusion of cisplatin may be a mechanism of drug resistance.

This work is clearly a technical "tour de force", as the authors show that srIBI is able to combine mass spectrometry imaging with super-resolution 3D imaging. The capacity to image a drug like cisplatin within cells without using any tags is an important achievement and can be extremely useful to better understand the biological action of drugs in general. Still, some of the claims made by the authors are overstated or lack strong experimental support (see below) and the manuscript is short on mechanistic insights regarding why cisplatin accumulates in nuclear speckles. However, beside these shortcomings, I think this work could be of interest for the readers of this journal if important questions are properly answered.

Major comments:

- Negative controls are always performed with no antibody, which does not reflect the real non-specific background of the MoC-Ab. It would be better to assess the true background of the MoC-Ab in cells depleted of the target protein by RNAi or gene KO. At least such control should be provided for one of the MoC-Ab.
- It is not clear if the authors performed their RNA-seq analysis using poly(A)⁺ RNA. This may underestimate the level of intron retention since pre-mRNAs are not all polyadenylated.
- While in figure 4D nuclear speckles are clearly visible in the two nuclei (in orange), these subnuclear domains seem to be gone in the analysis in figure 5. This is especially the case in figures 5E and 5H, where SC35 neighborhoods (6, 7 and 9) occupy most of the nucleoplasm, and nuclear speckles are not visible anymore. Furthermore, there is an increase in SC35 staining in cells treated with cisplatin compared to untreated cells (Figure S23E). Is there an increase in SC35 expression upon treatment with cisplatin in these cells? Is it possible that the SC35 staining with Streptavidin-Au197 is noisier in super-resolution srIBI imaging? Another possibility is that cisplatin treatment disrupts nuclear speckles, as reported before (Umehara et al., BBRC, 2003). Did the authors tried to image their cells treated with cisplatin by confocal microscopy to detect SC35-labeled nuclear speckles?
- Page 18, line 19: "srIBI observation that cisplatin accumulates within nuclear speckles". This is an overstatement since supporting data are coming from only 2 nuclei (figure 4E). Moreover, in Figure 5, cisplatin colocalized with SC35 but there is no evidence that nuclear speckles are present in these cells.
- Page 21, line 15-20: there are already several publications on the impact of cisplatin on mRNA

splicing (like Schmittgen et al., *Int. J. Oncol.*, 2003). This should be mentioned in the discussion.

- Figure 6C: labeling of the nucleus should be included in the cells displayed in this figure, so we could see more clearly if cisplatin is excluded or not from the nucleus.

- Figure 6E: the fraction of cisplatin positive voxels is clearly decreased in the nucleus of resistant cells. However, the cisplatin positive voxels do not increase in the cytoplasm (neighborhood 6), which should be expected if cisplatin was pumped out of the nucleus, as suggested by the authors. Another interpretation of their data is that cisplatin is pumped out from the cell, and the nucleus is more efficiently cleared out of the drug.

Reviewer #3:

Remarks to the Author:

In this work, the authors reported a new molecular imaging strategy, named "super-resolution ion beam imaging (srIBI)", which is based on isotopic labelling and secondary ion mass spectrometry imaging using NanoSIMS. Multi-components, including both small (e.g., cisplatin, an anti-cancer drug) and large (e.g., DNA, and protein) molecules can be imaged with very high spatial resolution (down to 30 nm), simultaneously. The idea is highly interesting but not very new, because I know quite a few research groups have been trying it. However, the data quality of this work is high (better than all works I have seen). The ion images are very beautiful and attractive. I love them! Therefore, I recommend it to be published in your journal, with some necessary revisions.

1. In this work, I saw isotopic-derivatized nucleotides were used (e.g., Fig S7, S8), so it should be some DNA/RNA molecules were imaged with anti-cancer drug molecules. However, the title is "Multiplexed PROTEIN and drug imaging by super-resolution ion beam imaging". Please revise it.

2. In page 2 line 3-4, "Current fluorescence-based microscopy methods for biomolecules in cells are constrained by the spatial resolution limitations, ...". However, it is well known that super resolution fluorescence can achieve 5-20 nm spatial resolution, which is better than 30 nm resolution that NanoSIMS can provide. Actually, the authors also mentioned in Page 24 line 5-6 "Super-resolution light microscopy methods are limited by the number of fluorophores that can be simultaneously recorded." Please fix this issue.

3. In page 7, line 2-4 "...down to 260 nm resolution 13. Attaining resolutions beyond this limit requires a beam source physically capable of forming a tighter beam". The Ref 13 was published in 2014. At that time, NanoSIMS' oxygen source was old duo-plasma ion source, so only 260 nm spatial resolution can be achieved. So far, a new ion source, called "hyper ion" has been available, and it can provide 40 nm spatial resolution (<http://www.oregon-physics.com/hyperion2.php>). Some new NanoSIMS instruments have been equipped with this new ion source. Therefore, negative secondary ion tag (such as F/Br/I) has less importance than before, because positive secondary ion tags (such as metal ion used in Cy-TOF) can be used to provide very decent spatial resolution, too. However, I know some small molecule drug, such as cisplatin, only sensitive in negative ion mode SIMS imaging. So negative ion tag is still very useful. Please add a few sentences to comment this issue.

4. In page 23, line 6-11, the author said that some new ion source was developed, which can provide very decent ultimate beam size (2 nm or better) for future SIMS imaging. However, the ultimate spatial resolution of SIMS is not limited by beam size, but some other factors. The cascade collision principle determines the ultimate resolution of SIMS to be around 10 nm (Barbara Garrison has some computer simulation 30+ years ago). Also, ionization yields and concentrations of the target molecules/atoms also play an important role. Please add a few sentences to comment this issue.

5. In page 23, line 14-15, "For example, with the seven-detector design presented here, a triple-barcode labeling would theoretically enable deconvolution of 35 targets." I am a little confused here. Please provide more details.

6. In Fig 1A, 3E, and 4B, microtome mode layer-by-layer 3D reconstruction was proposed. However, actual situation should be much more complex. Besides the topography issue, it is well

known that sputter rates of different locations of cells may be different. Therefore, to make a reliable 3D imaging reconstruction, ex situ/in situ AFM measurement has been used to correct z axle (e.g., Castner, *Anal. Chem.* 2012, 84(11), 4880–4885; Winograd, *Anal. Chem.* 2007, 79 (15), 5529-5539.). In page 23, line 15-20, the authors only used 1-2 sentences to address this issue. Please add more comments.

7. In the caption of the Fig 1F, "Quantification of the relationship between srIBI resolution and secondary ion yield." Obviously, "secondary ion yield" is not accurate here. I suggest using "secondary ion counts" to replace "secondary ion yield".

8. Fig 3F, the scale bar in the e-image: "4 mm"?? Should be 4 um, right?

9. Fig 5A, the scale bar in the first image: "2 mm"?? Should be 2 um, right?

10. In Fig 6C/D, the third row was labeled as "Merge". In SIMS field, "Overlay" has been more used.

Overall, in my opinion, this is a high-quality manuscript, and I love it. Therefore, I feel it can be accepted after some necessary revisions.

Reviewer #4:

Remarks to the Author:

The manuscript by Rovira-Clave and colleagues applies imaging mass spectrometry (nanosims) to capture antibody/chemical probes and cisplatin in the nucleus, which when performed in multiplexed fashion and coupled with multivariate statistical methods enables the identification of functionally relevant intranuclear nanodomains.

The group has previously used nanosims imaging of metal conjugated antibody probes to perform multiplexed characterization of tumor sections. The current study is analogous in the sense that antibody probes are being detected (metal or halogen conjugated) in multiplexed fashion, yet different in that they now seek to capture heterogeneous domains at the much smaller scale of an individual nucleus.

From an analytical standpoint, the advances presented are incremental relative to their prior work and when viewed with work from other groups, which collectively have demonstrated imaging of halogen conjugated nucleotides, metal-conjugated antibodies, halogen conjugated in situ probes, other heavy metal stains, etc. In some instances, such methods have been used to identify small intracellular structures and in the microbiology literature there are examples of nanosims imaging of bacteria of similar size to nucleoli or some of the other domains described in this manuscript (including after being probed with some of the aforementioned probes).

That being said, there are aspects of this manuscript that I think are important. While variations of the multivariate statistical methods that the authors' used to identify intranuclear domains are routinely used in single cell applications (Cytos, single cell RNA seq), their application to pixels on nanosims images is interesting. In addition, while it is easy for me to say that measurement of various antibody-based and other probes have been demonstrated with nanosims, the measurement of halogenated oligonucleotide-antibody conjugates of relevance to intranuclear biology, could open up new potential applications. In my opinion, however, the strength of the proof of concept biological conclusions in the latter portion of the manuscript are a weak point of the manuscript.

My specific points/questions are as follows:

1) Most (if not all) of the analyzed cells are whole mounted. SIMS is notoriously challenging for analysis of sample surfaces that are not flat. This is particularly important for direct measurement of individual ions where surface contours can affect ion yield, raising the question of how

consistent and generalizable the results are if applied across a larger number of cells or other cell types. While the nucleolus identification seems clear and robust (often visible in the e- images as well), would this issue not have the potential to cause identification of artifactual intranuclear domains and/or limit the capacity to quantify localization of molecules to these domains (e.g. cisplatin)?

2) Mixing of atoms (particularly in the z axis) as a result of Cesium deposition is often raised as a limitation with SIMS in general. Have the authors considered how that can affect their analyses and resolution? This may be particularly relevant "3D" analyses and their purported identification of nuclear speckles, which I believe are smaller than the reported resolution of nanosims.

3) Is cisplatin diffusible in the cell and how does sample processing affect distribution?

4) Regarding the experiment shown in Figure 6 and building on the Cisplatin mapping data in Figure 4: The differences in cisplatin distribution in the nuclear neighborhoods appear to be subtle at best as the distributions are overlapping with the exception of those at the periphery/nuclear membrane. Given my concern in point #1, this raises the question of whether the differences are real. Moreover, it is not clear to me that this is the best proof of concept of their intranuclear analytical and statistical mapping, as JQ1 lowers platinum signal in all of the intranuclear neighborhoods. Did we need this elaborate method to show that there was less platinum in the entire nucleus?

5) An additional question regarding the JQ1/Cisplatin treated cells: how do we know that they are true resistant cells as opposed to cells that are in early stages of cell death—an effect of dual treatment.

6) Why are all of the voxels merged from many different cells in the final figure 6E. It seems that this could be an opportunity to show the variability between cells, which is difficult to appreciate from this manuscript.

REVIEWER COMMENTS

We are grateful to the editor and reviewers for their time in assessing this work, as well as for the comments raised. We are delighted to share the excitement for this new technology, especially regarding the potential for biological studies and further methodological development. To address the reviewer's comments, we have carefully examined and provided a reply for each of the points raised. We appreciate the opportunity to improve the manuscript.

Reviewer #2 (Remarks to the Author):

In this manuscript, Rovira-Clave and colleagues report the development of super-resolution ion beam imaging (srlBI), a mass spectrometry-based imaging technique capable of high-parameter imaging in 3D. The authors describe in details the design and validation of mass-oligonucleotide-conjugated antibodies (MoC-Ab), which they use to detect and image specific proteins using srlBI. They also validated multiparametric imaging and the identification of subnuclear structures. The manuscript also describes the application of srlBI for 3D cellular imaging and reconstruction at nanoscale level.

The authors apply multiparametric srlBI to image cisplatin distribution in TYK-nu cells. They report an enrichment of cisplatin in nuclear speckles, which are sites of splicing factor accumulation, suggesting a link between cisplatin and mRNA splicing. RNA-seq analysis further suggest an increased intron retention in cells treated with cisplatin. Finally, srlBI imaging of cells resistant to cisplatin and the BET inhibitor JQ1 reveals that cisplatin is excluded from the nucleus of these cells, suggesting that nuclear exclusion of cisplatin may be a mechanism of drug resistance.

This work is clearly a technical "tour de force", as the authors show that srlBI is able to combine mass spectrometry imaging with super-resolution 3D imaging. The capacity to image a drug like cisplatin within cells without using any tags is an important achievement and can be extremely useful to better understand the biological action of drugs in general. Still, some of the claims made by the authors are overstated or lack strong experimental support (see below) and the manuscript is short on mechanistic insights regarding why cisplatin accumulates in nuclear speckles. However, beside these shortcomings, I think this work could be of interest for the readers of this journal if important questions are properly answered.

We appreciate the reviewer's time taken in reading this manuscript in detail, as well as the helpful feedback and comments.

Major comments:

- Negative controls are always performed with no antibody, which does not reflect the real non-specific background of the MoC-Ab. It would be better to assess the true background of the MoC-Ab in cells depleted of the target protein by RNAi or gene KO. At least such control should be provided for one of the MoC-Ab.

The reviewer raised an important point. The physical properties of antibodies might be affected by conjugation, since the disulphide bonds within the antibody are mildly disrupted and a negatively-charged adduct (the oligonucleotide) is coupled to them.

In the Nolan lab, we have successfully developed a set of protocols for oligo-antibody conjugations and have extensive experience in their applications to biological samples. We pioneered the effort with CODEX (Goltsev, Y., et al, Cell, 2018 and Schurch, C., et al, Cell, 2020), quantum barcoding (O'Huallachain, M. et al, Communications Biology, 2020) and now it is similarly applied to srIBI. It is also noteworthy that the oligo conjugation protocol is analogous to the original CyTOF conjugation (Bendall, S., et al, Science, 2011), which was also pioneered in the Nolan lab. A key feature of retaining antibody activity is through a controlled mild reduction step, followed by rapid conjugation to a deprotected polymer (a maleimide-oligo in this case). The oligo sequences are also designed to minimize cross-reactivity and non-specific binding, in addition to the role of salmon-sperm DNA used in the protocol.

A similar approach has also been utilized by a number of other labs in technologies including immuno-SABER (Saka, S., et al, Nature Biotechnology, 2019), CITE-seq (Stoeckius, M., et al, Nature Methods, 2017) and REAP-seq (Peterson, V.M., et al, Nature Biotechnology, 2017).

In the current manuscript we conjugated oligonucleotides to anti-sc35 (for nuclear speckles), anti-CENP-A (for centromeres), anti-nucleolin, anti-fibrillarin and anti-nucleophosmin (for the nucleolus), anti-dsDNA (for DNA), anti-h3k27ac (open chromatin), anti-h3k27me3 (close chromatin) and anti-TOMM20 (for mitochondria). All srIBI experiments resulted in expected and distinctive patterns, which are comparable to the same antibodies before conjugation.

The suggestion for depleting a target protein through knockdown or knockout experiments is for **early** validation for **new** antibodies. These are well characterized reagents. We have hundreds of experiments of historical controls. If the same standard were held for every study involving pre-validated antibodies, experiments would be unwieldy. Knock downs are we believe unnecessary and would be outside the scope of this manuscript.

- It is not clear if the authors performed their RNA-seq analysis using poly(A)+ RNA. This may underestimate the level of intron retention since pre-mRNAs are not all polyadenylated.

The RNA-seq analysis was performed using poly(A)+ RNA, as described in the materials and methods. To clarify the performed experiment in the main text, the original sentence in the Results section (page 18, lines 13 to 14):

“We inferred incomplete splicing (as determined by the presence of unspliced intronic sequences) from triplicate bulk RNA-seq experiments at two time points (Fig. 5I),”

has been modified to:

“We inferred incomplete splicing (as determined by the presence of unspliced intronic sequences) from triplicate bulk poly(A)+ RNA-seq experiments at two time points (Fig. 5I),”

We agree with the reviewer that our analysis might be an underestimate. A similar approach is also implemented in the commonly used RNA velocity methodology (La Manno et al, Nature, 2019). Nevertheless, we demonstrate that cisplatin interferes with appropriate processing of RNA by showing a significant increase of immature intronic sequences in poly(A)+ RNA from cisplatin treated cells. While the level of intron retention may be slightly underestimated due to a portion of unpolyadenylated pre-mRNAs, the observation is both statistically significant and striking even with this underestimation. Further detailed studies into the specific genes affected will be outside of the scope of this manuscript.

- While in figure 4D nuclear speckles are clearly visible in the two nuclei (in orange), these subnuclear domains seem to be gone in the analysis in figure 5. This is especially the case in figures 5E and 5H, where SC35 neighborhoods (6, 7 and 9) occupy most of the nucleoplasm, and nuclear speckles are not visible anymore. Furthermore, there is an increase in SC35 staining in cells treated with cisplatin compared to untreated cells (Figure S23E). Is there an increase in SC35 expression upon treatment with cisplatin in these cells? Is it possible that the SC35 staining with Streptavidin-Au197 is noisier in super-resolution sRII imaging? Another possibility is that cisplatin treatment disrupts nuclear speckles, as reported before (Umehara et al., BBRC, 2003). Did the authors tried to image their cells treated with cisplatin by confocal microscopy to detect SC35-labeled nuclear speckles?

We thank the reviewer for this astute observation. The cells in both Figure 4 and 5 underwent identical cisplatin treatment, and thus it is unlikely that the observed differences are modulated by cisplatin.

The images shown in Figures 5E and 5H are cartoon representations of the sampled data (20000 voxels), coloring clusters across the entire Z stack in a flattened 2D visualization. As such, some of the “speckles” may be lost due to the collapsing of the Z plane (although the information was retained for the neighborhood interaction analysis in 5F). Similarly, the raw images shown in 5A is also summed on the Z plane.

In addition, given selected field-of-views of cells were acquired for iterative-srIBI, it likely that some details were not captured in great detail. The reviewer is referred to Figure S24C for some nuclear speckles that are visible.

- Page 18, line 19: “srIBI observation that cisplatin accumulates within nuclear speckles”. This is an overstatement since supporting data are coming from only 2 nuclei (figure 4E). Moreover, in Figure 5, cisplatin colocalized with SC35 but there is no evidence that nuclear speckles are present in these cells.

We thank the reviewer for raising this concern. As eluded to, our data for Figure 5 is a summary across 20000 voxels sampled across 5 FOVs, in which only 1 (in collapsed Z depth) is shown as a representative view. In the others, seen in Figure S24C, nuclear speckles are present.

Of note, the analysis is also performed differently in Figure 4 (cisplatin signals was not used for voxel clustering) than in Figure 5 (cisplatin was used for voxel clustering). From the neighborhood interaction frequency data in Figure 5F, SC35 clusters (6, 7 and 8) so show a high propensity of aggregating together, akin to the nuclear speckle features that are expected.

- Page 21, line 15-20: there are already several publications on the impact of cisplatin on mRNA splicing (like Schmittgen et al., Int. J. Oncol., 2003). This should be mentioned in the discussion.

We thank the reviewer for pointing this out.

We have modified the following sentence in the Discussion section of the manuscript (page 21, lines 16 to 17):

“Complemented with a decreased mRNA splicing on cisplatin treated cells as showed by global transcriptome analysis,”

to read:

“Complemented with a decreased mRNA splicing on cisplatin treated cells as showed by our global transcriptome analysis and work from others,”

- Figure 6C: labeling of the nucleus should be included in the cells displayed in this figure, so we could see more clearly if cisplatin is excluded or not from the nucleus.

We thank the reviewer for this comment. Since the nuclear structures of the cells were clearly visible through the secondary electron images (Figs 6B and 6C), we attempted to reduce a visual overload on the readers through another series of images. Additional unbiased quantification of the data in 6E supports the conclusion for cisplatin exclusion from the nucleus.

- Figure 6E: the fraction of cisplatin positive voxels is clearly decreased in the nucleus of resistant cells. However, the cisplatin positive voxels do not increase in the cytoplasm (neighborhood 6), which should be expected if cisplatin was pumped out of the nucleus, as suggested by the authors. Another interpretation of their data is that cisplatin is pumped out from the cell, and the nucleus is more efficiently cleared out of the drug.

We thank the reviewer for this comment and agree on this additional interpretation. We have modified the following sentence in the Discussion section of the manuscript (page 22, lines 4 to 8):

“Interestingly multi-drug resistance to cisplatin and other drugs with surface receptors is a well-documented phenomenon and begs the question of whether other classes of small molecule pumps have evolved to maintain the nucleus free from certain metabolites or environmental poisons.”

to read:

“Interestingly multi-drug resistance to cisplatin and other drugs with surface receptors is a well-documented phenomenon and begs the question of whether other classes of small molecule pumps have evolved to maintain the nucleus free from certain metabolites or environmental poisons, mechanisms that might act in concert for a more efficient pumping out of drugs from the cell.”

Reviewer #3 (Remarks to the Author):

In this work, the authors reported a new molecular imaging strategy, named “super-resolution ion beam imaging (srIBI)”, which is based on isotopic labelling and secondary ion mass spectrometry imaging using NanoSIMS. Multi-components, including both small (e.g., cisplatin, an anti-cancer drug) and large (e.g., DNA, and protein) molecules can be imaged with very high spatial resolution (down to 30 nm), simultaneously. The idea is highly interesting but not very new, because I know quite a few research groups have been trying it. However, the data quality of this work is high (better than all works I have seen). The ion images are very beautiful and attractive. I love them! Therefore, I recommend it to be published in your journal, with some necessary revisions.

We thank the reviewer for the positive assessment of our technology and share in the excitement for the potential of the method. We address the concerns raised below.

1. In this work, I saw isotopic-derivatized nucleotides were used (e.g., Fig S7, S8), so it should be some DNA/RNA molecules were imaged with anti-cancer drug molecules. However, the title is “Multiplexed PROTEIN and drug imaging by super-resolution ion beam imaging”. Please revise it.

We thank the reviewer for this insightful comment, as we realize that the original title might be confusing to the reader. In this manuscript, we used isotope-derivatized nucleotides to synthesize single-stranded oligonucleotides that were conjugated to antibodies. Therefore:

1. When a signal is recorded, it is because the isotope-derivatized nucleotides on the oligonucleotide were detected.
2. The spatial location of the signal is determined by the antibody recognizing its antigen.

As such, the oligonucleotide may be the “tag” that registers the ion counts on the mass-spectrometer, but the antibody, which targets proteins of interest in this study, is the functional molecule that localizes the oligonucleotides to the specific proteins of interest. This methodology has been used by a number of labs, including ours, in antibody-oligo conjugates for protein detection, including CODEX (Goltsev, Y., et al, Cell, 2018 and Schurch, C., et al, Cell, 2020), quantum barcoding (O’Huallachain, M. et al, Communications Biology, 2020), immuno-SABER (Saka, S., et al, Nature Biotechnology, 2019), CITE-seq (Stoeckius, M., et al, Nature Methods, 2017) and REAP-seq (Peterson, V.M., et al, Nature Biotechnology 2017).

To better reflect this concept, the original title:

“Multiplexed protein and drug imaging by super-resolution ion beam imaging”

has been modified to:

“Subcellular localization of biomolecules and drug distribution by super-resolution ion beam imaging”.

2. In page 2 line 3-4, “Current fluorescence-based microscopy methods for biomolecules in cells are constrained by the spatial resolution limitations, ...”. However, it is well known that super resolution fluorescence can achieve 5-20 nm spatial resolution, which is better than 30 nm resolution that NanoSIMS can provide. Actually, the authors also mentioned in Page 24 line 5-6 “Super-resolution light microscopy methods are limited by the number of fluorophores that can be simultaneously recorded.” Please fix this issue.

We thank the reviewer for this comment. We have now removed the sentence “*Current fluorescence-based microscopy methods for biomolecules in cells are constrained by the spatial resolution limitations, parameters per experiment, and detector systems*” from the abstract. The sentence in Page 24, lines 20-21:

“Super-resolution light microscopy methods are limited by the number of fluorophores that can be simultaneously recorded”

has been modified to:

“Super-resolution light microscopy of multiple biomolecules simultaneously on the same sample is challenging”.

3. In page 7, line 2-4 “...down to 260 nm resolution 13. Attaining resolutions beyond this limit requires a beam source physically capable of forming a tighter beam”. The Ref 13 was published in 2014. At that time, NanoSIMS’ oxygen source was old duo-plasma ion source, so only 260 nm spatial resolution can be achieved. So far, a new ion source, called “hyper ion” has been available, and it can provide 40 nm spatial resolution (<http://www.oregon-physics.com/hyperion2.php>). Some new NanoSIMS instruments have been equipped with this new ion source. Therefore, negative secondary ion tag (such as F/Br/I) has less importance than before, because positive secondary ion tags (such as metal ion used in Cy-TOF) can be used to provide very decent spatial resolution, too. However, I know some small molecule drug, such as cisplatin, only sensitive in negative ion mode SIMS imaging. So negative ion tag is still very useful. Please add a few sentences to comment this issue.

The reviewer is right that the new Hyperion II ion source has better imaging resolution than the duoplasmatron, theoretically enabling a spatial resolution similar to the one shown in this study by using lanthanide-conjugated antibodies (used in CyTOF, MIBI and IMC). In fact, the Hyperion technology has since been implemented on the MIBIScope

through collaborations between Oregon Physics and Ionpath (of which G.P.N. is a cofounder).

If we limit the scope of the current work to the increase in imaging resolution, we agree with the reviewer that the usefulness of negative secondary ions detection might be undermined by this newly developed gun. However, the cesium gun not only provides high spatial resolution, it also enables detection of negative secondary ions (as acknowledged by the reviewer). In other words, the tools developed in this study are better compatible with simultaneous detection of stable isotopes, such as ^{13}C and ^{15}N . This sets the stage for future work studying the subcellular distribution of stable isotope alternatives of most drugs and other small-molecules (lipids, metabolites, etc.) while mapping cellular landmarks with antibodies.

Along the same lines, Helium and Neon sources, coupled with SIMS (on the ORION HIM-SIMS instrument) can attain sub 10-nm resolutions, and is discussed in the manuscript:

“Other primary beam sources, such as helium- or neon-based beams, which have beam sizes of 0.5 nm and 2 nm, respectively, have been recently adapted for SIMS⁶⁷”

That said, we have added to the Discussion section of the manuscript (page 23, lines 18 to 20) the following sentence:

“If negative secondary ion detection is not required, the new Hyperion II ion source installed in some NanoSIMS provides lateral resolution down to 40 nm and it is compatible with lanthanide-conjugated antibodies”.

4. In page 23, line 6-11, the author said that some new ion source was developed, which can provide very decent ultimate beam size (2 nm or better) for future SIMS imaging. However, the ultimate spatial resolution of SIMS is not limited by beam size, but some other factors. The cascade collision principle determines the ultimate resolution of SIMS to be around 10 nm (Barbara Garrison has some computer simulation 30+ years ago). Also, ionization yields and concentrations of the target molecules/atoms also play an important role. Please add a few sentences to comment this issue.

Indeed, the ionization efficiency for each element is another key driver for SIMS, in addition to beam size/resolution determinants. The following sentence has been added to comment on this issue (page 23, lines 10 to 14):

“Other primary beam sources, such as helium- or neon-based beams, which have beam sizes of 0.5 nm and 2 nm, respectively, have been recently adapted for SIMS⁶⁷. Successful implementation of these technologies for effective targeted imaging, through methods akin to s/IBI, may require additional signal amplification and chemistry modifications to achieve more efficient ionization efficiencies.”

5. In page 23, line 14-15, “For example, with the seven-detector design presented here, a triple-barcoded labeling would theoretically enable deconvolution of 35 targets.” I am a little confused here. Please provide more details.

We thank the reviewer to provide us with the opportunity to elaborate on this concept.

The NanoSIMS 50L has a multicollection system consisting of 7 detectors. In this manuscript, we leverage on fluorine, bromine and iodine to identify the spatial distribution of biomolecules within cell neighborhoods through their covalent link to antibodies (MoC-Abs). Selenium and tellurium are two other element that can be efficiently ionized by the cesium beam and have been previously linked to nucleic acids (Jiang, J. et al, Nucleic Acids Research, 2007 and Sheng, J. et al, Nucleic Acids Research, 2007, respectively). Altogether, these 5 elements present with 12 stable isotopes abundant enough to synthesize MoC-Abs.

In the manuscript, we rationalize that improvements in resolution (e.g., physical expansion of samples, improvements in beam precision, etc.) might enable 3D imaging of single antibodies, which have an approximate size of 10 nm in diameter.

Assuming single antibody detection, combinatorial approaches might provide to the 7 detectors design an increase on the absolute number of detected biomolecules. For example, the isotopes ^{19}F , ^{76}Se , ^{79}Br , ^{81}Br , ^{122}Te , ^{125}Te and ^{127}I could be leveraged for the synthesis of nucleoside phosphoramidites. These reagents would be combined in groups of three to generate 35 unique mass-oligonucleotides (**Figure R1**). Each mass-oligonucleotide could be conjugated to a distinct antibody. Despite other more complicated strategies could be leveraged for barcoding, an n-choose-k would ensure that only regions in 3D space with three of the seven isotopes will be selected as real signal.

That said, we have added a supplementary note (**Supplementary Note 2**) related to the sentence “For example, with the seven-detector design presented here, a triple-barcoded labeling would theoretically enable deconvolution of 35 targets (**Supplementary Note 2**).” in the discussion section of the manuscript (page 23, lines 17 to 18).

Figure R1. Schematic representation of isotope barcoding. Antibodies would be conjugated to mass-oligonucleotides synthesized with three of the seven isotope-derivatized nucleotides for a total of 35 combinations.

Of additional interest is our recent publication on this barcoding concept on the NanoSIMS (Harmsen et al, Adv Mater Technol, 2020).

6. In Fig 1A, 3E, and 4B, microtome mode layer-by-layer 3D reconstruction was proposed. However, actual situation should be much more complex. Besides the topography issue, it is well known that sputter rates of different locations of cells may be different. Therefore,

to make a reliable 3D imaging reconstruction, ex situ/in situ AFM measurement has been used to correct z axle (e.g., Castner, Anal. Chem. 2012, 84(11), 4880–4885; Winograd, Anal. Chem. 2007, 79 (15), 5529-5539.). In page 23, line 15-20, the authors only used 1-2 sentences to address this issue. Please add more comments.

We thank the reviewer for this comment and the opportunity to address it adequately. We have modified the following paragraph in the Discussion section of the manuscript (page 23, lines 21 to 23 and page 24, lines 1 to 7):

“Analysis of samples with heterogeneous atomic density and uneven surfaces, such as the cells shown in this work, result in potential for distortions and differential sputtering yields through the etching process that affect 3D reconstruction. Instrument improvements, embedding of registration agents such as nanorods, applications of atomic force microscopy or photogrammetric 3D reconstruction, hold promise to correct for such potential “artefacts”.”

to read:

“Analysis of samples with uneven topography and presenting locations with heterogeneous atomic density, such as the cells shown in this work, result in potential for distortions and differential sputtering yields through the etching process that affect 3D reconstruction. Atomic force microscopy (AFM) is a powerful method to correct for such distortions by providing a snapshot of the sample topography before and after the analysis. However, artefacts arising from sample transfer and distinct chamber environments are still a limitation. In situ AFM, the integration of AFM with SIMS instruments, would be the ideal theoretical solution but it is currently hindered by instrumental availability and technical challenges. Instrument improvements, embedding of registration agents such as nanorods, or photogrammetric 3D reconstruction, all hold promise for future exploration.”

7. In the caption of the Fig 1F, “Quantification of the relationship between srIBI resolution and secondary ion yield.” Obviously, “secondary ion yield” is not accurate here. I suggest using “secondary ion counts” to replace “secondary ion yield”.

We thank the reviewer for pointing this out. We have substituted “secondary ion yield” by “secondary ion counts” in the revised version of the manuscript.

8. Fig 3F, the scale bar in the e-image: “4 mm”?? Should be 4 um, right?

We thank the reviewer for pointing at this mistake. We have corrected it in the revised version of the manuscript.

9. Fig 5A, the scale bar in the first image: “2 mm”?? Should be 2 μm , right?

We thank again the reviewer for pointing at this mistake. We have corrected it in the revised version of the manuscript.

10. In Fig 6C/D, the third row was labeled as “Merge”. In SIMS field, “Overlay” has been more used.

We thank the reviewer for this suggestion. We have substituted “Merge” by “Overlay” in the revised version of the manuscript.

Overall, in my opinion, this is a high-quality manuscript, and I love it. Therefore, I feel it can be accepted after some necessary revisions.

We are delighted to share the reviewer’s thoughts and excitement, and are very thankful for the time put into careful reading of this manuscript.

Reviewer #4 (Remarks to the Author):

The manuscript by Rovira-Clave and colleagues applies imaging mass spectrometry (nanosims) to capture antibody/chemical probes and cisplatin in the nucleus, which when performed in multiplexed fashion and coupled with multivariate statistical methods enables the identification of functionally relevant intranuclear nanodomains.

The group has previously used nanosims imaging of metal conjugated antibody probes to perform multiplexed characterization of tumor sections. The current study is analogous in the sense that antibody probes are being detected (metal or halogen conjugated) in multiplexed fashion, yet different in that they now seek to capture heterogeneous domains at the much smaller scale of an individual nucleus.

From an analytical standpoint, the advances presented are incremental relative to their prior work and when viewed with work from other groups, which collectively have demonstrated imaging of halogen conjugated nucleotides, metal-conjugated antibodies, halogen conjugated in situ probes, other heavy metal stains, etc. In some instances, such methods have been used to identify small intracellular structures and in the microbiology literature there are examples of nanosims imaging of bacteria of similar size to nucleoli or some of the other domains described in this manuscript (including after being probed with some of the aforementioned probes).

That being said, there are aspects of this manuscript that I think are important. While variations of the multivariate statistical methods that the authors' used to identify intranuclear domains are routinely used in single cell applications (Cytos, single cell RNA seq), their application to pixels on nanosims images is interesting. In addition, while it is easy for me to say that measurement of various antibody-based and other probes have been demonstrated with nanosims, the measurement of halogenated oligonucleotide-antibody conjugates of relevance to intranuclear biology, could open up new potential applications. In my opinion, however, the strength of the proof of concept biological conclusions in the latter portion of the manuscript are a weak point of the manuscript.

We thank the reviewer for the thorough assessment of the manuscript and the time invested. The comments are valid and we are grateful for pushing us to ensure critical limitations are well discussed. While it is true that individual tags (such as for bacteria or antibodies labeled with ^{19}F or ^{197}Au) have been individually performed in a number of separate studies before, they have not been combined at the scale in which we perform this study (due to the tagging of the oligonucleotide on the antibody instead of directly labeling the antibody itself).

My specific points/questions are as follows:

1) Most (if not all) of the analyzed cells are whole mounted. SIMS is notoriously challenging for analysis of sample surfaces that are not flat. This is particularly important for direct measurement of individual ions where surface contours can affect ion yield, raising the question of how consistent and generalizable the results are if applied across a larger number of cells or other cell types. While the nucleolus identification seems clear and robust (often visible in the e- images as well), would this issue not have the potential to cause identification of artifactual intranuclear domains and/or limit the capacity to quantify localization of molecules to these domains (e.g. cisplatin)?

The reviewer raises an astute point, uneven topography is challenging. Despite this issue, we provide evidence that the staining pattern for each antibody is as expected when compared to fluorescence microscopy. As a multiparameter imaging platform, each signal also acts as counterstain or proxy for expected structures (e.g., in figure S15, CENP-A on the edges of the nucleolus is expected as proxy for the acrocentric chromosomes centromeres). We successfully validated our antibodies to identify their respective nuclear subdomains, both after image acquisition as well as unsupervised identification of nuclear neighborhoods during the analysis, indicating the data generated is robust.

The unsupervised stratification of nuclear and cytoplasmic regions across 22 individual cells (Fig. 6 and S31) across 4 independent treatment conditions also demonstrate the robustness of both the methodology and analytical framework.

Because the topography challenge is of relevance to the reader, we have extended the Discussion section of the manuscript to read (page 23, lines 21 to 23 and page 24, lines 1 to 7):

“Analysis of samples with uneven topography and presenting locations with heterogeneous atomic density, such as the cells shown in this work, result in potential for distortions and differential sputtering yields through the etching process that affect 3D reconstruction. Atomic force microscopy (AFM) is a powerful method to correct for such distortions by providing a snapshot of the sample topography before and after the analysis. However, artefacts arising from sample transfer and distinct chamber environments are still a limitation. In situ AFM, the integration of AFM with SIMS instruments, would be the ideal theoretical solution but it is currently hindered by instrumental availability and technical challenges. Instrument improvements, embedding of registration agents such as nanorods, or photogrammetric 3D reconstruction, all hold promise for future exploration.”

2) Mixing of atoms (particularly in the z axis) as a result of Cesium deposition is often raised as a limitation with SIMS in general. Have the authors considered how that can affect their analyses and resolution? This may be particularly relevant “3D” analyses and their purported identification of nuclear speckles, which I believe are smaller than the reported resolution of nanosims.

The mixing of atoms into each z depth is certainly a key physical limitation of SIMS in general. To minimize the potential confounders experimentally, we routinely presputter the samples for at least a minute on high current to reach steady state of deposition, as well as relative uniformity of sample depth.

The analysis pipeline was developed to understand the *relative* positions of voxels of data, as defined by us in a 3 x 3 x 10 pixel space, and account for two matters:

1. The potential noise introduced by deposition of ions from the layer above, and
2. The relative position of each voxel to the voxels immediately surrounding it in 3D space.

These considerations allow us to maximize the true signal to noise ratios, while circumventing any potential Z depth artefacts as voxel position is relative to those around it, and not absolutely fixed in 3D space.

Given that the sizes of nuclear speckles can range from a few hundreds of nanometers to the micron scale (Kim, J. et al, J Cell Sci, 2019), they are well above the current resolution limit of nanoSIMS. The same holds true for other nuclear domains identified in this manuscript.

3) Is cisplatin diffusible in the cell and how does sample processing affect distribution?

The reviewer raises an excellent question. Cisplatin is a small molecule that enters the cell through protein pumps and passive diffusion. Its intracellular distribution is expected to occur through passive diffusion, with contribution of intracellular pumps for its distribution within membranous organelles being currently unknown. Leveraging imaging means might provide a better understanding of such potential mechanisms, as demonstrated in this manuscript.

Free cisplatin is small enough to diffuse freely in the cell, where it can be incorporated into DNA, RNA and proteins into DNA/RNA-adducts as well as protein crosslinks. Since our staining protocol includes extensive washes after chemical fixation, it is most likely we are washing away unbound cisplatin, but retaining cisplatin covalently linked to biomolecules such as DNA, RNA and proteins.

To discuss this, we have included a sentence in the Discussion section of the manuscript (page 24, lines 7 to 11):

“Cisplatin has been observed to form adducts with DNA and RNA, while forming crosslinks to proteins. After sample processing, unbound cisplatin is washed away but covalently-linked cisplatin remains. Cryosectioning together with genetically-encoded chemical tags can provide a workflow for comparative studies into unbound versus bound intracellular drug distribution.”

4) Regarding the experiment shown in Figure 6 and building on the Cisplatin mapping data in Figure 4: The differences in cisplatin distribution in the nuclear neighborhoods appear to be subtle at best as the distributions are overlapping with the exception of those at the periphery/nuclear membrane. Given my concern in point #1, this raises the question of whether the differences are real. Moreover, it is not clear to me that this is the best proof of concept of their intranuclear analytical and statistical mapping, as JQ1 lowers platinum signal in all of the intranuclear neighborhoods. Did we need this elaborate method to show that there was less platinum in the entire nucleus?

We thank the reviewer for providing the opportunity to elaborate on details and relevance of the method. Exactly to the reviewer’s point, we set out initially to see if cisplatin was excluded specifically from any particular nuclear neighborhoods but did not find any statistically differences (or visual, as pointed out by the reviewer) (Fig. 6E). Notably, the resolution for the data acquired in Fig 6 was also lower (Table. S4) than Fig. 4 to accommodate for the amount of time required to image the larger number of cells. As such, the lower imaging resolution can have an impact on some of the analysis for nuclear neighborhoods, but does not deviate away from the robustness of using our voxel-based approach, since we can clearly separate out cytoplasm and nuclear neighborhoods with differentially enrichment of the nuclear markers (Fig. 6A) with visual confirmation (Fig. 6B and S31).

Building on our observation that cisplatin is enriched in certain nuclear neighborhoods (per citation of our BioRxiv preprint posted in 2019), a recent manuscript observes DNA-cisplatin adducts, or chemically modified cisplatin-fluorophore, accumulates in specific nuclear protein condensates *in vitro* (Klein, I. A., et al, Science, 2020). The addition of JQ1 to their cisplatin-treated cells, high analogous to our experiments and data, also observes a lower abundance of DNA-cisplatin adducts in the nucleus when compared to cisplatin alone. These results highlight the need to advance our understanding of subcellular drug localization even with the chemical and biological tools currently available.

It would be incorrect to posit “One could just label the small molecule with fluorescence” since that would in fact be altering the molecule and leave us subject to the criticism the effect is driven by the alteration itself. To better illustrate this point, we incubated TYK-nu cells with three small fluorescent molecules (fluorescein, sulforhodamine 101 acid chloride (Texas Red) and 5-carboxytetramethylrhodamine (5-TAMRA)) and assessed

their subcellular distribution. Fluorescein was homogeneously distributed through the nucleus and cytoplasm in a majority of cells but highly enriched in the cytoplasm of certain cells (**Figure R2A**). Texas Red presented with a punctate staining in the cytoplasm (**Figure R2B**). 5-TAMRA showed a similar distribution than Texas Red with some cells additionally presenting extensive circular accumulations of the fluorophore in the cytoplasm (**Figure R2C**). Altogether, these results show a distinct subcellular distribution among three commonly used small molecules. If a small drug is conjugated to one of these fluorescent molecules... Who would drive the subcellular distribution of such complex?

The reviewer argues other methods could be leveraged to reach the same conclusion. Yes, if you already knew the result, it's fine to say "I could have done this another way". But it is beyond unfair to imply that an approach used to discover something doesn't have merit just because it might have been accomplished some other way. And yet, despite 50 years of studying this key drug... no one did discover it.

Nearly any discovery can be discerned by other techniques... if you already know what you are looking for. There is no technique that allows for visualization and localization of an unlabeled drug to subcellular compartments... especially when you do not know the compartments ahead of time. To criticism our approach and efforts is highly biased as the end result is now known (and therefore fairly obvious).

Biomolecular imaging is extremely useful because it can inform mechanism without previous assumptions. While protein and nucleic acid biologists have taken advantage of such tool for decades, to obtain our results using less elaborate methods and without a *priori* knowledge, all the following requirements need to be met:

- identification of the small molecule without changes to its chemical structure
- spatial location at the nanometer scale
- identification of landmarks within the cell to which to ascertain potential mechanisms

To the best of our knowledge, srIBI is the only mature method with such abilities. Despite its current limitations (they are discussed in the manuscript and their targeting is expected to be a fruitful arena), our method sets foundation, and there will certainly be better ways to do that in the future.

The main objective of this work was to develop a method for high resolution visualization of small molecules within molecularly relevant subcellular locations, provide a generalizable analytical framework for such type of data and exemplify how leveraging on this method could provide answers to questions previously difficult to target. We believe we have succeeded on that.

Figure R2. Small fluorescent molecules are unevenly distributed through cells. Representative images of TYK-nu cells incubated with 10 μ M of fluorescein (A), sulforhodamine 101 acid chloride (Texas Red) (B) or 5-carboxytetramethylrhodamine (5-TAMRA) (C) for 1 hour. Images at the bottom are an enlarged version of the red dashed boxes of the images on top.

5) An additional question regarding the JQ1/Cisplatin treated cells: how do we know that they are true resistant cells as opposed to cells that are in early stages of cell death—an effect of dual treatment.

We thank the reviewer for this comment. JQ1 and cisplatin dual resistant cells have been previously deeply characterized in an ovarian cell culture model and mouse system (Yokoyama, Y., et al, Cancer Res., 2016). In our ovarian cancer cell lines, we performed dual drug treatments at similar concentrations and multiple timepoints. To avoid potential secondary mutations due to the reported rapid acquisition of resistance by ovarian cancer cells, we only let the experiment run for 72 hours as ovarian cancer cells have a high potential of acquiring cisplatin resistance rapidly (Parker, et al, JCI, 1991).

6) Why are all of the voxels merged from many different cells in the final figure 6E. It seems that this could be an opportunity to show the variability between cells, which is difficult to appreciate from this manuscript.

We thank the reviewer for pointing out the potential for heterogeneity even at the subcellular scale. The data in Fig. 6E represented the distributions of each individual voxel, from which 5000 voxels were sampled from each of the cells (10 cells that were Cisplatin and 9 that were Cisplatin + JQ1). These voxels were not merged together, but rather maintained their individuality for statistical power.

We also would like to point out that nuclear neighborhood distribution (Fig. S30F) and spatial distribution of the cells (Fig. S31) were both not visibly variable between the two conditions tested here, and hence were not the purpose of the study.

Reviewers' Comments:

Reviewer #2:

Remarks to the Author:

The authors have properly answered the main comments of the reviewers. I think this work is an important achievement and the revised manuscript is now acceptable for publication in this journal.

Reviewer #3:

Remarks to the Author:

The authors already addressed all my concerns, and I feel the manuscript can be published now. Also, thanks for the detailed explanation for the 35 possibilities.

Only a minor issue: in Page 24, line 518, "embedding of registration agents such as nanorods". I agree that "embedding of registration agents" should be a good idea. However, nanorods may have significantly different sputter rates if compared to cell species, and it may lead to some extra issues in 3d NanoSIMS analysis. Therefore, please remove "such as nanorods".

By the way, as a SIMS scientist, I am very excited to see the development of MIBI and relevant extensions. I hope it can be called MIBI-SIMS. However, a name is less important than the science. I must say: "I love MIBI"!

Reviewer #4:

Remarks to the Author:

The authors' have provided a detailed rebuttal. Although I continue to like the paper, for full disclosure, in my initial review of the manuscript, I thought it was borderline for publication in Nature Communications. While they have addressed some of the concerns with modifications to the text, they have not engaged my concerns experimentally. I will briefly focus on the original points that I made that I believe remain inadequately addressed by textual modifications (points 1, 4, and 5) corresponding to the numbering from my original review. I also make some comments regarding point 6, which is related to their presentation/description of the data, for their consideration.

1) I would have liked to see some experimental data to address the question of topology effects. They referenced AFM which could indeed provide some indication of the degree to which topology modified counts of relevant ions. One could also imagine testing their method out in samples that did not have topology issues (e.g. sectioned cells). I appreciate their view on this issue--and to be clear I am not suggesting that all of their results are artifactual and due to topology. However, in my opinion some experimental exploration of the degree to which topology affects their results is warranted.

4) In my original comment, I was referring not to the NanoSIMS analysis, but as I stated to the "intranuclear analytical and statistical mapping." I apologize if this was unclear. The more traditional approach to capturing data for intracellular domains from NanoSIMS images, would be to select easily identifiable structures such as the nucleus or nucleolus or cytoplasm (as done for cisplatin imaging by the referenced paper, Lee et al Metallomics 2017), where it was shown that there were differences in cisplatin penetration in resistant cells. If the authors applied this method (or simply visualized the cells), it seems likely that there is nucleus-wide exclusion of the platinum in the JQ1/Cisplatin treated cells. As such it is perhaps a forgone conclusion that they would also see this reflected in subnuclear domains as revealed by their more sophisticated statistical analysis. For this reason, I continue to view this particular analysis as a suboptimal proof of concept for their approach. I

5) I don't believe that the authors can assume that the dynamics of cell death and resistance as shown by others in this cell line are identical in their specific passage and in their hands. This seems to be of some importance: as the authors note, cisplatin is transported into the cell, and therefore it may require a living cell to achieve the requisite concentrations in the cell (and nucleus). So an alternative hypothesis to explain their finding is that the lack of platinum is due to dead or dying cells. They reference their phase contrast images in Figure S29, which is insufficient evidence. Moreover, are the cisplatin mono therapy treated cells not also resistant to cisplatin??

6) In my original review, I considered this point to be relatively minor. However, I remain somewhat confused by the authors' response and hope that my comments will help them reconsider their presentation of the data or at least provide additional information in the figure legend as to exactly what these distributions represent. Am I wrong in viewing Figure 6E as having merged the voxels from 10 and 9 cells respectively into a single distribution? Meaning cisplatin treated group $N=5000 \times 10=50K$ and Cis/JQ1 $N=5000 \times 9=45K$? If this is true, then it would be helpful to know how much of this effect was driven by a minority of cells or if the effect was consistent across cells.

REVIEWER COMMENTS

Reviewer #2 (Remarks to the Author):

The authors have properly answered the main comments of the reviewers. I think this work is an important achievement and the revised manuscript is now acceptable for publication in this journal.

We thank the reviewer for the help in getting this manuscript improved.

Reviewer #3 (Remarks to the Author):

The authors already addressed all my concerns, and I feel the manuscript can be published now. Also, thanks for the detailed explanation for the 35 possibilities. Only a minor issue: in Page 24, line 518, "embedding of registration agents such as nanorods". I agree that "embedding of registration agents" should be a good idea. However, nanorods may have significantly different sputter rates if compared to cell species, and it may lead to some extra issues in 3d NanoSIMS analysis. Therefore, please remove "such as nanorods".

By the way, as a SIMS scientist, I am very excited to see the development of MIBI and relevant extensions. I hope it can be called MIBI-SIMS. However, a name is less important than the science. I must say: "I love MIBI"!

We would like to thank the reviewer again for improving this manuscript. As requested, we have removed "such as nanorods" from the main text.

Reviewer #4 (Remarks to the Author):

The authors' have provided a detailed rebuttal. Although I continue to like the paper, for full disclosure, in my initial review of the manuscript, I thought it was borderline for publication in Nature Communications. While they have addressed some of the concerns with modifications to the text, they have not engaged my concerns experimentally. I will briefly focus on the original points that I made that I believe remain inadequately addressed by textual modifications (points 1, 4, and 5) corresponding to the numbering from my original review. I also make some comments regarding point 6, which is related to their presentation/description of the data, for their consideration.

We thank the reviewer for his/her comments and clarifications. We believe our replies will sufficiently address the concerns raised.

1) I would have liked to see some experimental data to address the question of topology effects. They referenced AFM which could indeed provide some indication of the degree to which topology modified counts of relevant ions. One could also imagine testing their method out in samples that did not have topology issues (e.g. sectioned cells). I appreciate their view on this issue--and to be clear I am not suggesting that all of their results are artifactual and due to topology. However, in my opinion some experimental exploration of the degree to which topology affects their results is warranted.

To address the reviewer's concerns regarding whether topology effects interfere with our method, we refer the reviewer to the tissue section analysis described in **Figure S10B** (included below for convenience). These results demonstrate that srlBI is robust in a number of different sample preparation methods and is not hindered by topology effects to the best of our knowledge.

As an alternative to the nucleolus reconstruction, we also include here a 3D reconstruction of CENP-A-stained centromeres (**Figure R1**, related to Figure S18D), which are relatively small substructures (~250-300 nm) that cannot be identified in SIMS secondary electron images without specific molecular targeting. Our data are in agreement with CENP-A staining results using SMLM (Andronov, et al., 2019, Nature Communications) and exemplifies how specific subcellular structures at a scale of a few hundreds of nanometers can be identified by srlBI (even in samples with uneven topography).

We resonate with the reviewer on the importance of topography. In fact, we acknowledge in the discussion the topography effects. Specifically, we say: "*Analysis of samples with uneven topography and presenting locations with heterogeneous atomic density, such as the cells shown in this work, result in potential for distortions and differential sputtering yields through the etching process that affect 3D reconstruction*". We then propose a series of alternatives to ameliorate this issue, which are significant endeavors beyond the scope of this manuscript. In our opinion, AFM is the most promising alternative but it presents with its own set of challenges. As the reviewer hopefully appreciates, combining srlBI and AFM is a major undertaking outside of the scope of this paper, such as building a unique instrument as exemplified by Wirtz, et al., 2013 Surf. Interface Analy. (this instrument has since been taken apart from our personal communications with Dr. Tom Wirtz). The objective of this manuscript was to demonstrate that high resolution and multiplexed molecular targeting can be achieved by SIMS, and to exemplify how this method might extend the unique SIMS capabilities to measure intrinsic (e.g., platinum in Cisplatin) and targeted biomolecules simultaneously. We have provided an extensive set of controls and an elaborated discussion about the method's limitations, and encourage further research by the SIMS field on this strategy.

B
Figure S10. Validation of MoC-Abs in frozen tissue sections. (B) Representative srIBI images of the epidermis of a human adult skin section stained with anti-H3K9me3-⁸¹Br/Cy3. Scale bar, 4 μm.

Figure R1. 3D surface reconstruction of CENP-A signal. HeLa cells were stained with anti-CENP-A-⁸¹Br/Cy3, and 40 individual planes were acquired to obtain srIBI images of centromeres from its appearance to its disappearance. 3D surface reconstruction of images of CENP-A reveal centromeres with the expected shape.

4) In my original comment, I was referring not to the NanoSIMS analysis, but as I stated to the "intranuclear analytical and statistical mapping." I apologize if this was unclear. The more traditional approach to capturing data for intracellular domains from NanoSIMS images, would be to select easily identifiable structures such as the nucleus or nucleolus or cytoplasm (as done for cisplatin imaging by the referenced paper, Lee et al Metallomics 2017), where it was shown that there were differences in cisplatin penetration in resistant cells. If the authors applied this method (or simply visualized the cells), it seems likely that there is nucleus-wide exclusion of the platinum in the JQ1/Cisplatin treated cells. As such it is perhaps a forgone conclusion that they would also see this reflected in subnuclear domains as revealed by their more sophisticated statistical analysis. For this reason, I continue to view this particular analysis as a suboptimal proof of concept for their approach.

The reviewer appears to refer to our analytical framework developed in Figures 4 and 5, stating it would not be appropriate for Figure 6 as we did not find any statistically significant differences between cisplatin concentrations in the various nuclear neighborhoods.

We respectfully disagree with this for the following reasons:

1. The statistical framework developed, combined with the targeted methodology, enable the identification of differential enrichment of cisplatin in the nuclear speckles (Figures 4 and 5).
2. The same framework, when applied to JQ1+Cisplatin results, showed depletion of Cisplatin in ALL nuclear neighborhoods identified. The reviewer is incorrectly discounting this, stating we could have "simply visualized the cells". This is simply untrue as we would not be able to quantify cisplatin differences in:
 - i) Heterochromatin
 - ii) Euchromatin
 - iii) Nuclear Speckles

As these structures cannot neither be "simply visualized" nor qualify for "intranuclear analytical and statistical mapping" without both the srlBI experimental and other analytical frameworks (see also Figure R1 above)—including staining with labelled antibodies. Combining visualization of these structures through targeted antibodies, together with unmodified drugs, is a feat impossible with previous studies (e.g., Lee et al).

5) I don't believe that the authors can assume that the dynamics of cell death and resistance as shown by others in this cell line are identical in their specific passage and in their hands. This seems to be of some importance: as the authors note, cisplatin is

transported into the cell, and therefore it may require a living cell to achieve the requisite concentrations in the cell (and nucleus). So an alternative hypothesis to explain their finding is that the lack of platinum is due to dead or dying cells. They reference their phase contrast images in Figure S29, which is insufficient evidence. Moreover, are the cisplatin mono therapy treated cells not also resistant to cisplatin??

Here, the reviewer is implying that the absence of nuclear-Cisplatin signal in the JQ1+Cisplatin resistant cells is due to them being dead or dying. We strongly disagree with this interpretation for the following reasons:

1. Cisplatin is known to strongly bind nonspecifically to dead but not live cells. This is the basis for a number of cisplatin-based live/dead assays developed in our lab (Bodenmiller et al 2012 Nature Biotechnology, Fienberg et al 2013 Cytometry A), which is extensively used for live/dead cell identification or dead-cell barcoding via CyTOF (a common multiparameter single-cell analytical method). Those results are commercialized and used as fact by hundreds of labs using mass spectrometry-based mass cytometry. Specifically, the reagent we used, Cell-ID Cisplatin (Fluidigm), has the following description summarized from all the relevant research:

"Cisplatin binds covalently to cellular proteins and labels cells with compromised cell membranes to a much greater extent than live cells. Therefore, Cell-ID Cisplatin specifically identifies dead cells when incubated prior to fixation or identifies total cells when incubated following cell fixation and permeabilization."

JQ1 can induce cell death in a number of cancer cell lines, including TYK-nu cells. To further demonstrate this, we incubated control and JQ1-treated TYK-nu cells with cisplatin for 1 minute, before CyTOF analysis to quantify the amount of cisplatin present in each cell. The number of cisplatin positive events was higher in the JQ1 sample compared to the control, indicative of higher levels of cell death. This result is expected due to JQ1-induced cell death (**Figure R2**). This result exemplifies that cisplatin strongly binds to dead cells, indicative the dual drug resistant cells are alive due to the absence of cisplatin in their nuclei.

Figure R2. Cisplatin binds to dead cells.

Representative CyTOF analysis of dead cells in TYK-nu ovarian cancer cells after cisplatin staining. TYK-nu cells were treated with 2.5 uM JQ1 for 4 days to induce cell death, incubated with 25 uM cisplatin for 1 minute, washed, and analyzed by CyTOF.

2. As the reviewer points out, "therefore it may require a living cell to achieve the requisite concentrations in the cell". Indeed, for a clear exclusion event instead of homogenous cisplatin signal in the cell, the cell would have to be a living cell.

Drug resistance, particularly in cancer cells, is a well-documented phenomenon. The combination of JQ1 and cisplatin induces cell death in our TYK-nu cultures, but certainly a number of resistant cells survive. These cells are alive, as demonstrated by trypan blue staining followed by quantification of live cells after 3 days of the dual treatment (**Figure R3**). These data are further evidence for dual drug resistant TYK-nu cells, which are the subject of our srlBI analysis.

Figure R3. TYK-nu cells can resist the double JQ1 and cisplatin treatment. TYK-nu cells were treated with JQ1 (2.5 μ M) or cisplatin (15 μ M) for 3 days, washed, cultured for 3 days, and counted using trypan blue to quantify the number of alive cells in the culture ($n = 3$). Images on the right are representative of cultured cells post treatment.

6) In my original review, I considered this point to be relatively minor. However, I remain somewhat confused by the authors' response and hope that my comments will help them reconsider their presentation of the data or at least provide additional information in the figure legend as to exactly what these distributions represent. Am I wrong in viewing Figure 6E as having merged the voxels from 10 and 9 cells respectively into a single distribution? Meaning cisplatin treated group $N=5000 \times 10=50K$ and Cis/JQ1 $N=5000 \times 9=45K$? If this is true, then it would be helpful to know how much of this effect was driven by a minority of cells or if the effect was consistent across cells.

We thank the reviewer for clarifying his/her comments, and allowing us to better explain our data.

Figure 6E was generated by calculating the fraction of voxels that were cisplatin positive across each cell under each condition, as noted in the Figure legends and the Y axis. As such, there is a single value per cell (Fraction of cisplatin-positive voxels, aka #Cisplatin-Positive Voxels / #Total Voxels), and 19 values for each neighborhood (10 cells treated with Cisplatin only, 9 cells treated with JQ1+Cisplatin).

To “know how much of this effect was driven by a minority of cells or if the effect was consistent across cells”, we present an additional plot below (**Figure R4**). Our conclusions remain unchanged: that cisplatin was actively excluded from all identifiable nuclear domains but not from the cytoplasm.

We would like to point out that the single-cell quantification plot was showed in Figure S34B of the original manuscript as a Cumming plot (included below for convenience), to show the difference in distribution as well as on the single cells as requested by the reviewer.

We have now added the following sentence to the legend of figure 6E:

“See Figure S34B for a representation of the variability on each single cell.”

Figure R4. Each dot represents the mean cisplatin across cisplatin-positive voxels for a cell. Colored red are the mean cisplatin for Cisplatin treated individual cells, and in green are the JQ1 + Cisplatin treated individual cells.

Figure S34. Neighborhood composition statistics. (B) The fraction of sampled voxels from each of the 6 annotated neighborhoods identified in Fig. 5A that were positive for cisplatin counts are plotted in a Cumming plot, showing the mean differences and 95% confidence intervals of JQ1 + cisplatin compared to cisplatin alone. Each dot is a cell.

Reviewers' Comments:

Reviewer #3:

Remarks to the Author:

Reviewer #4's comment #1 and authors' reply

My comment: I feel there may be some misunderstanding between the reviewer and the authors. The reviewer's point seems: this work cannot be "directly" called 3D imaging because it is not a "perfect" 3D reconstruction due to the topography issue and sputter rate variation among different locations/components. Therefore, the authors' tone is too high, though this work itself is very nice. Meanwhile, the authors claim that this is "definitely" a 3D imaging work, though with some weaknesses. I suggest the authors (1) use "Pseudo 3D imaging" in this work to lower their tone a little, (2) say this is only an initial try, and (3) say 3D reconstruction using AFM help or "sectioned cells" will be used in their future work.

Reviewer #4's comment #4 and authors' reply

My comment: Though the authors' reply is somewhat reasonable, I agree more with the reviewer #4's comment. Especially, I also tend to "view this particular analysis as a suboptimal proof of concept for their approach". So, if possible, please lower the tone a little. Of course, both the reviewer #4 and I feel this is a very nice work.

Reviewer #4's comment #5 and authors' reply

My comment: I tend to support the authors' reply. My background is SIMS, and I am not an expert in cancer cell – anti-cancer drug research, though I did use NanoSIMS and ToF-SIMS to help some collaborators in this direction. So, I am not sure if I am a suitable person to judge this debate. However, as my current experience, I tend to support the authors' reply, because they did publish quite a few high-quality papers in this field.

Reviewer #4's comment #6 and authors' reply

My comment: The authors agreed with the reviewer's comment. They gave some explanations in the reply letter and did some corresponding revisions in the manuscript. I feel both the comment and the reply are reasonable.

REVIEWERS' COMMENTS

Reviewer #3 commenting on Reviewer #4's comments (Remarks to the Author):

We thank Reviewer #3 and Reviewer #4 for the help in getting this manuscript improved.

Reviewer #4's comment #1 and authors' reply

My comment: I feel there may be some misunderstanding between the reviewer and the authors. The reviewer's point seems: this work cannot be "directly" called 3D imaging because it is not a "perfect" 3D reconstruction due to the topography issue and sputter rate variation among different locations/components. Therefore, the authors' tone is too high, though this work itself is very nice. Meanwhile, the authors claim that this is "definitely" a 3D imaging work, though with some weaknesses. I suggest the authors (1) use "Pseudo 3D imaging" in this work to lower their tone a little, (2) say this is only an initial try, and (3) say 3D reconstruction using AFM help or "sectioned cells" will be used in their future work.

We thank the reviewer for these suggestions.

The sentence in the results sections (Page 6, line 3):

"HD-MIBI uses a positively charged cesium primary ion beam with a small spot size to obtain 3D super-resolution multiparametric visualization of cellular features (Fig. 1A), including the distribution of small molecules."

has been modified to:

"HD-MIBI uses a positively charged cesium primary ion beam with a small spot size to obtain pseudo 3D super-resolution multiparametric visualization of cellular features (Fig. 1A), including the distribution of small molecules."

The sentence in the results sections (Page 12, line 3):

"3D reconstruction at the nanoscale with HD-MIBI"

has been modified to:

"Pseudo 3D reconstruction at the nanoscale with HD-MIBI"

The sentence in the results sections (Page 12, line 3):

"To demonstrate the feasibility of whole organelle 3D reconstruction, we acquired 785 planes of a single nucleolus (Fig. 3A and Fig. S19) and performed a volumetric reconstruction (Fig. 3B and Movies S1 and S2)."

has been modified to:

“As a preliminary experiment, and to demonstrate the feasibility of whole organelle pseudo 3D reconstruction, we acquired 785 planes of a single nucleolus (Fig. 3A and Fig. S19) and performed a volumetric reconstruction (Fig. 3B and Movies S1 and S2).”

A paragraph in the discussion section of the revised manuscript (page 24, lines 1 to 10) extensively describes the current limitations of 3D imaging and future work:

“Analysis of samples with uneven topography and presenting locations with heterogeneous atomic density, such as the cells shown in this work, result in potential for distortions and differential sputtering yields through the etching process that affect 3D reconstruction. Atomic force microscopy (AFM) is a powerful method to correct for such distortions by providing a snapshot of the sample topography before and after the analysis^{70,71}. However, artefacts arising from sample transfer and distinct chamber environments are still a limitation. *In situ* AFM, the integration of AFM with SIMS instruments, would be the ideal theoretical solution but it is currently hindered by instrumental availability and technical challenges⁷². Instrument improvements, reconstruction of serially sectioned cells, or photogrammetric 3D reconstruction⁷³, also hold promise for future exploration.”

As suggested by the reviewer, we have included the sentence “reconstruction of serially sectioned cells” (page 24, line 9).

Reviewer #4's comment #4 and authors' reply

My comment: Though the authors' reply is somewhat reasonable, I agree more with the reviewer #4's comment. Especially, I also tend to “view this particular analysis as a suboptimal proof of concept for their approach”. So, if possible, please lower the tone a little. Of course, both the reviewer #4 and I feel this is a very nice work.

We thank the reviewer for this comment. To lower the tone, we have modified the following sentence in the Discussion section of the manuscript (page 26, line 8 to 10):

“Future improvements should include better methods to extract data meaningfully.”

to read:

“Future improvements should include better methods to extract data meaningfully, including their combination with conventional analysis.”

Reviewer #4's comment #5 and authors' reply

My comment: I tend to support the authors' reply. My background is SIMS, and I am not an expert in cancer cell – anti-cancer drug research, though I did use NanoSIMS and

ToF-SIMS to help some collaborators in this direction. So, I am not sure if I am a suitable person to judge this debate. However, as my current experience, I tend to support the authors' reply, because they did publish quite a few high-quality papers in this field.

We thank the reviewer for this comment.

Reviewer #4's comment #6 and authors' reply

My comment: The authors agreed with the reviewer's comment. They gave some explanations in the reply letter and did some corresponding revisions in the manuscript. I feel both the comment and the reply are reasonable.

We thank the reviewer for this comment.